# LET SSMs BE CONVNETS: STATE-SPACE MODELING WITH OPTIMAL TENSOR CONTRACTIONS

**Yan Ru Pei**
Brainchip Inc.
Laguna Hills, CA 92653, USA
`yanrpei@gmail.com`

## ABSTRACT

We introduce Centaurus, a class of networks composed of generalized state-space model (SSM) blocks, where the SSM operations can be treated as tensor contractions during training. The optimal order of tensor contractions can then be systematically determined for every SSM block to maximize training efficiency. This allows more flexibility in designing SSM blocks beyond the depthwise-separable configuration commonly implemented. The new design choices will take inspiration from classical convolutional blocks including group convolutions, full convolutions, and bottleneck blocks. We architect the Centaurus network with a mixture of these blocks, to balance between network size and performance, as well as memory and computational efficiency during both training and inference. We show that this heterogeneous network design outperforms its homogeneous counterparts in raw audio processing tasks including keyword spotting, speech denoising, and automatic speech recognition (ASR). For ASR, Centaurus is the first network with competitive performance that can be made fully state-space based, without using any nonlinear recurrence (LSTMs), explicit convolutions (CNNs), or (surrogate) attention mechanism.

## 1 INTRODUCTION

Sequence or temporal modeling encompasses a wide range of tasks from audio processing to language modeling. Traditionally, there have been many (related) statistical methods employed (Box et al., 2015). In the age of deep learning, neural networks have been predominantly used (LeCun et al., 2015), including recurrent neural networks (RNNs), convolutional neural networks (CNNs), transformers (Vaswani, 2017), and neural ODEs (Chen et al., 2018). In many cases, the model will inevitably suffer from one of two drawbacks: 1) cannot be efficiently trained (or fitted) in parallel due to the sequential nature of the model, 2) cannot be efficiently configured for online inference due to its large memory and computational requirement. To address this, deep state-space models (SSMs) were adapted for sequence modeling, and have shown incredible potential across a wide range of tasks (Gu et al., 2021; Goel et al., 2022; Gu & Dao, 2023). Due to the linearity of the SSM layers, they can not only be configured for efficient online inference with small memory and computational resources, but also configured for efficient training using parallel hardware with unrolling strategies (Gu et al., 2022; Smith et al., 2022; Dao & Gu, 2024; Heinsen, 2023).

Currently, most deep SSM networks (along with most neural networks in general) follow the architectural recipe of transformers, where they are composed of uniform "SSM blocks" throughout the network, containing little to no variations in the shapes of the intermediate features or weights. This simplifies the designs of deep SSM networks, but may sacrifice performance and efficiency in practice. To explore the opposite direction, we go "back to the future" to classical CNN designs instead, where a much more heterogeneous design principle is followed. More specifically, as we go deeper in the network, we will gradually downsample the temporal dimension, increase the channel dimension, and increase connective sparsity (Tan & Le, 2021), to balance between size and efficiency (Pei & Coenen, 2024). In order to allow for the possibility of such a heterogeneous design, we make two novel contributions in this work. First, we express SSM blocks as tensor networks (or with `einsum` expressions), where we can easily observe the connective structure of existing SSM blocks as being mostly depthwise-separable. This motivates us to design new SSM blocks using this

tensor network formalism, giving us connective structures such as full and bottleneck SSM blocks, as inspired by classical CNN blocks. Then, we optimize the contraction order of each SSM block dynamically based on the type of the SSM block, the shape of the input features, and the shapes of the SSM system matrices. This enables significant speedups during training for all SSM blocks (both old and new ones). We name this class of efficient CNN-like networks Centaurus[1]. The model source code is at github.com/Brainchip-Inc/Centaurus.

## 2 RELATED WORK

### 2.1 DEEP STATE-SPACE MODELING

The seminal work proposing a memory encoding using orthogonal Legendre polynomials in a recurrent state-space model is the Legendre Memory Unit (LMU) (Voelker et al., 2019), where Legendre polynomials (a special case of Jacobi polynomials) are used. The HiPPO formalism (Gu et al., 2020) then generalized this to other common orthogonal functions. Later, this sparked a cornucopia of works interfacing with deep state-space models including S4 (Gu et al., 2021), H3 (Fu et al., 2022), and Mamba (Gu & Dao, 2023), achieving impressive results on a wide range of tasks from audio generation Goel et al. (2022) to language modeling. Besides a few recent exceptions (Smith et al., 2022; Dao & Gu, 2024), These networks mostly use an underlying depthwise structure, which may limit the network capacity, albeit reducing the compute requirement of the network. An important focal point of this work is to generalize such SSM models to enable more connective flexibility, such that design choices of classical convolutional blocks can be carried over to the Centaurus model.

### 2.2 CLASSICAL CONVOLUTIONAL BLOCKS

Here, we look at the variants besides the standard full convolutional layer (where every pair of input and output channels is connected). First, the simplest variant is the depthwise convolutional layer, made popular by the MobileNetV1 architecture (Howard et al., 2017). This variant connects the input and output channels one-to-one, which drastically reduces the connectivity of the architecture, hence also its parameters and computational complexity. This architecture is prevalent for both computer vision and speech domains (Kriman et al., 2020; Hannun et al., 2019). Next, a variant with moderate connectivity is the grouped convolutional layer, appearing as early as the AlexNet work (Krizhevsky et al., 2012). In this structure, the input/output channels are divided into groups, and each input-channel group is associated with an output-channel group one-to-one. The input and output channels are then only intra-connected within each group, but there are no inter-connections between groups. Finally, there is a class of convolutional blocks known as bottlenecks, typically containing a sequence of three convolutional layers. This structure is used prominently in the ResNet model (He et al., 2016), and also in MobileNetV2 (Sandler et al., 2018) onwards. Typically, the order of convolutional layers can be pointwise-depthwise-pointwise, resulting in an output feature of the same tensor shape as the input feature.

### 2.3 TENSOR NETWORKS

Tensor networks were originally developed as a tool for approximating the wavefunctions of many-body quantum systems (Orús, 2019). It can be considered as an incredibly generalized form of low-rank decomposition, where high-dimensional features can be compressed as the contraction of a few low-dimensional tensors (Kolda & Bader, 2009). It has a direct relationship to the einsum expression, where each einsum operand is considered a node of the tensor network, and each contraction index is considered a (hyper)edge. The memory- and compute-optimal contraction or evaluation of a tensor network can be formulated as a congestion optimization problem on the base graph, allowing tensor networks to contest with "quantum supremacy" (Gray & Kourtis, 2021). The application of tensor networks to machine learning began with the seminal work of Novikov et al. (2015), where it is shown that the fully connected layers of a neural network can be exponentially

---

[1]As a quick explanation of the name, we are exploring Tensorized Temporal Networks, or TenTeNs, from which we get a hundred, or CENTaurus. Admittedly, Centaurus is not actually composed of the word root "cent", so this wordplay is not too sensible. However, Centaurus is a constellation, usually visualized as stars being linked; this is visually similar to what a tensor network looks like, so some naming sense is recovered.

compressed with minimal degradation in performance. This spawned a series of works applying tensor networks for compressing convolutional and transformer blocks. Our work extends this line of "tensorization" technique to deep state-space models.

# 3 BACKGROUND

## 3.1 STATE SPACE MODELS

State-space models (SSMs) are general representations of linear time-invariant (LTI) systems (Hamilton, 1994), and they can be uniquely specified by four matrices: $A \in \mathbb{R}^{N \times N}$, $B \in \mathbb{R}^{N \times H}$, $C \in \mathbb{R}^{H' \times N}$, and $D \in \mathbb{R}^{H' \times H}$. The first-order ODE describing the LTI system is given as

$$\dot{x} = Ax + Bu, \qquad y = Cx + Du, \tag{1}$$

where $u \in \mathbb{R}^H$ is the input signal, $x \in \mathbb{R}^N$ is the internal state, and $y \in \mathbb{R}^{H'}$ is the output. The parameters $\{H, H', N\}$ denote the number of features for the input, output, and internal state respectively. Setting $H = H' = 1$ yields a single-input, single-output (SISO) SSMs (Gu et al., 2021; 2022), and letting $H > 1$, $H' > 1$ yields a multiple-input, multiple-output (MIMO) SSMs (Smith et al., 2022). We discretize the system using zero-order hold (ZOH), which gives us the discrete-time SSM matrices $\overline{A}$ and $\overline{B}$ as follows (Gu et al., 2022):

$$\overline{A} = \exp(\Delta A), \qquad \overline{B} = (\Delta A)^{-1} \cdot (\exp(\Delta A) - 1) \cdot \Delta B. \tag{2}$$

The discrete SSM is then given by

$$x[t+1] = \overline{A}\,x[t] + \overline{B}\,u[t], \qquad y[t] = C\,x[t] \tag{3}$$

It is then straightforward to check that the discrete-time impulse response is given as $k[\tau] = C\,\overline{A}^\tau\,\overline{B}$, where $\tau$ denotes the kernel timestep. During training, $k$ can be considered the "full" long 1D convolutional kernel with shape (output channels, input channels, length), in the sense that the output $y$ can be computed via the long convolution $y_j[t] = \sum_i (u_i * k_{ji})[t]$.

Similar to previous works (Gu et al., 2022), we assume $\overline{A}$ to be complex diagonal, but restrict $\overline{B}$ and $C$ to be real projection matrices to reduce memory and computational loads. In addition, we ignore the term $Du$ as it can be absorbed into the SSM system. Justification of these restrictions is given in Appendix A. Like previous works also, we allow the parameters $\{A, B, C, \Delta\}$ to be directly learnable, which indirectly trains the kernel $k$. Unlike previous works in deep SSMs, we do not try to keep the sizes $H$, $H'$, $N$ consistent, to allow for more flexibility in feature extraction at each layer, mirroring the flexibility in selecting channel and kernel sizes in CNNs[2]. The flexibility of the tensor shapes requires a careful choice of the optimal order of operations during training to minimize memory and computation, which will be the focal point of this work.

## 3.2 EINSTEIN SUMMATION NOTATION

The Einstein summation notation (Einstein, 1922), or `einsum`, is a concise representation of general tensor contractions. We do not give a formal description of this notation here, but instead introduce it in the context of describing a MIMO SSM. Recall that the output of a MIMO SSM is computed by convolving the input with the impulse response, which we normally would write as

$$y_j[t] = \sum_i (u_i * k_{ji})[t] = \sum_i \left( u_i * \left( \sum_n \overline{B}_{ni} K_n C_{jn} \right) \right)[t], \tag{4}$$

where we defined the basis kernels $K[\tau] = \Re(\overline{A}^\tau)$. We get a rather messy expression involving three summation indices. $i$ denotes the input channel, $n$ denotes the internal state index, and $j$ denotes the output channel.

To lessen the notation burden, we observe that the summation expression is redundant and can be inferred from the tensor indices. In particular, if an index appears on the RHS of the equation but

---

[2]The size of the internal state $h$, can be interpreted as the degree of parametrization of a basis temporal kernel, or some implicit (dilated) "kernel size" in the frequency domain.

not the LHS, then it must have been summed over (or contracted). This allows us to reduce the expression down to

$$y_j[t] = \left( u_i * \left( \overline{B}_{ni} K_n C_{jn} \right) \right)[t], \qquad (5)$$

which is much better than before, and we can always assume the indices $i$ and $n$ to be summed over without the explicit hinting of a summation symbol. We can further simplify equation 5 by discarding convolution operator $*$. Naturally, we can leverage the convolution theorem, which states that the convolution operator is mapped to pointwise multiplication in the frequency domain. If we index the Fourier modes using $f$, then equation 5 can be expressed in the frequency domain as

$$\hat{y}_j[f] = \hat{u}_i[f] \overline{B}_{ni} \hat{K}_n[f] C_{jn} \quad \text{or} \quad \hat{y}_{jf} = \hat{u}_{if} \overline{B}_{ni} \hat{K}_{nf} C_{jn} \qquad (6)$$

where the hat symbol denotes the Fourier transformed features (see Appendix A for details).

### 3.3 A GENERALIZATION

Looking at $K_n$, we realize that there is only one oscillation mode per internal coefficient $n$, which in certain cases may limit the expressiveness of the network. A natural generalization is to expand the basis kernels as $K_{nm}$, and arrive at the following system that is more expressive:

$$\hat{y}_{jf} = \hat{u}_{if} \overline{B}_{ni} \hat{K}_{nmf} E_{nm} C_{jn}. \qquad (7)$$

Here, $E_{nm}$ serves as additional weighting factors for the basis kernels (which again we restrict to be real), representing the importance of each oscillation mode. The recurrent form of this system is then

$$u'_n[t] = \overline{B}_{ni} u_i[t], \qquad x_{nm}[t+1] = \overline{A}_{nm} x_{nm}[t] + u'_n, \qquad y_j[t] = C_{jn} E_{nm} \Re(x_{nm}[t]), \qquad (8)$$

which can be considered a parallel cascade of $n$ SISO state-space systems, intra-connected by the $E_{nm}$ factors, and inter-connected by the $\overline{B}_{ni}$ and $C_{jn}$ projection matrices. Alternatively, we can consider $n$ to index a "state block", and $m$ to index "sub-states" within the state block, which is an extension beyond the pure SISO and MIMO configurations. As a sidenote, this configuration is similar to the Mamba block architecture (Gu & Dao, 2023), but without the data-gating mechanism and the gated MLP structure.

Like Mamba, it is possible to configure the SSM matrices to be data-dependent: $A(u)$, $B(u)$, $C(u)$, $D(u)$, in which case the system becomes time-variant (Katsch, 2023; Gu & Dao, 2023; Dao & Gu, 2024). We will not place a major focus on this configuration in our work for two (temporary) reasons. Under the present algorithmic understanding, such dynamic systems require materialization of the internal states for scan operations (in Mamba 1) or require the introduction of a new sequence dimension into the tensor operands (in Mamba 2), meaning that it can restrict the flexibility of tensor contraction orders. On the more practical end, such models generally require custom kernels (in Triton or CUDA) for specialized support currently, making it difficult to build heterogeneous networks with different connective configurations. We believe however that there are no fundamental restrictions to combining our framework with the data-gating mechanism in theory, and leave it as a future direction of study to achieve this efficiently in practice.

## 4 SSMs AND CNNs ARE TENSOR NETWORKS

Within an SSM layer, the temporal kernels are constructed via the basis kernels, which are further generated (or parameterized) by the recurrent coefficients in $A$. In other words, a temporal kernel can be generated by a weighted sum of selected Fourier modes, where each mode $n$ is associated with a complex frequency of $A_n$ and a (real) weighting factor $E_n$. A natural viewpoint is that the parameters of $A$ and $E$ are analogous to the parameters of convolutional filters, but not directly parameterized. For instance, if we consider $A$ having 3 complex parameters and $E$ having 3 real parameters, then they together may contain the same "expressivity" as a $3 \times 3$ spatial filter, both representable with 9 real numbers. One may note that standard CNN spatial filters are local in space, while our temporal IIR filters are global in time. Interestingly however, by themselves, the difference between local convolutions and global convolutions is not too fundamental. This is because, by the convolution theorem, a temporal convolution is just simply a pointwise product in the frequency domain, and vice versa. Therefore, a dual viewpoint is that in the frequency domain, deep SSM models are simply pointwise operations with "non-local" activation functions in between playing the "frequency mixing" role. This idea is also explored in the Hyena work (Poli et al., 2023).

### 4.1 GENERAL SSM OPERATIONS WITH EINSUM EXPRESSIONS

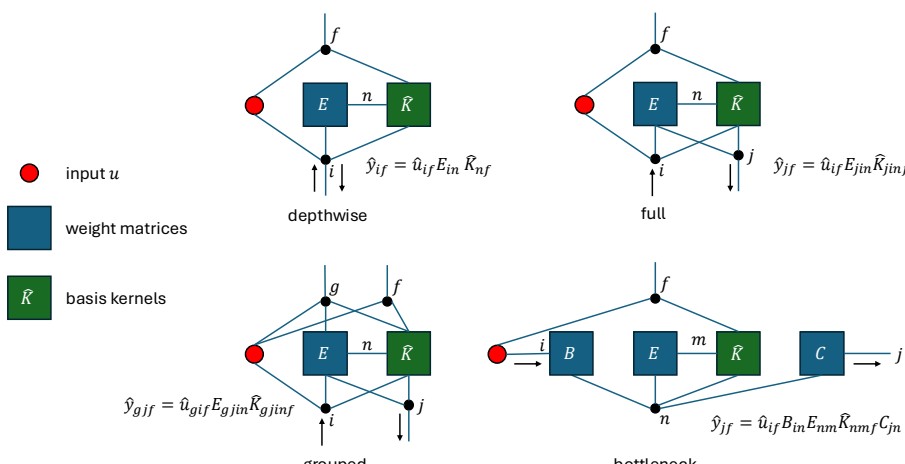

Figure 1: A tensor network representation of the tensorial connection structure of the different Centaurus block configurations, inspired by classical convolutional blocks. Here, the indices $\{i, j, n, g, f\}$ index the input channel, output channel, internal state, group/head, and Fourier mode respectively. Each (hyper)edge is denoted with one of the indices, representing the dimension to contract over. Note that the hyperedge associated with $f$ always connects to three tensors, as it represents a convolution in the temporal domain, involving two input tensors and one output tensor.

In standard CNN layers, besides the canonical convolution operation, there is usually an additional interaction between the input and output channels. For instance, we have configurations such as "depthwise convolution", "group convolution", and "full convolution", in increasing degree of channel connectivity. In this section, we will detail how to use the einsum expression to endow the $A$ tensor with these channel-interaction structures. Naturally, matrices that are purely channel-mixing like $B$ and $C$ are akin to "pointwise convolution" layers or simply projection operations, which we will place less emphasis on.

From here on, we will use $\{i, j, n, g, f\}$ to index the input channel, output channel, internal state, group/head, and Fourier mode respectively. The simplest example is a "depthwise" SSM block, or $\hat{y}_{if} = \hat{u}_{if} E_{in} \hat{K}_{in}$, noting the appearance of the "input channel" index $i$ in every operand and the total absence of the "output channel" index $j$. Therefore, the layer by itself will not see any interactions among the "channel" dimension, hence why an additional mixing matrix $M$ is needed at the end $\hat{y}'_{jf} = \hat{y}_{if} M_{ji}$. In short, we have a pure sequence-mixing layer followed by a pure channel-mixing layer, forming a depthwise-separable structure, as also discussed in Gu et al. (2022).

On the other extreme, we have a "full" SSM block, or $\hat{y}_{jf} = \hat{u}_{if} E_{jin} \hat{K}_{jin}$, where both indices $i$ and $j$ appear for the basis kernels, as each input-output pairing needs to be separately parameterized. In this case, both the sequence-mixing and channel-mixing structures are "baked" into the full SSM block, so not additional mixing layers are necessary. Figure. 1 depicts a tensor network representation of the different connectivity structures corresponding to classical CNN blocks.

### 4.2 OPTIMAL CONTRACTION ORDERS FOR TRAINING: THE DEPTHWISE EXAMPLE

In online inference mode, we have no choice but to explicitly materialize and evolve the internal states, leaving little room for optimization of the order of operations. This is also true for other parallelization strategies where the internal states are materialized in some form (Smith et al., 2022; Heinsen, 2023). Fortunately, we have the flexibility of choosing the order of contraction if we perform the SSM operations in the frequency domain (e.g. via FFTs), and considerable speedups (at times, orders of magnitudes) can be achieved by choosing the optimal contraction order. A slight drawback of using FFT is that the time complexity is $O(L \log L)$ instead of the desired $O(L)$.

However, this is typically not a practical issue, as FFT is rarely a compute-bound operation unless the sequence length is exceedingly large or the operation is already kernel-fused (Fu et al., 2022). At this point of discussion, we will start accounting for the batch dimension as $b$, omitted previously for clarity of exposition. In practice, the batch size will factor into the determination of the optimal contraction order.

Taking again the depthwise SSM layer, or $\hat{y}_{bif} = \hat{u}_{bif} E_{ni} \hat{K}_{nif}$, intuitively we would opt to contract the $E$ and $\hat{K}$ tensors first to "collapse" down the basis kernels along the state dimension (indexed $n$) as such, $\hat{k}_{if} = E_{ni} \hat{K}_{nif}$. It then becomes much more manageable to compute the output by simply taking the product $\hat{y}_{bif} = \hat{u}_{bif} \hat{k}_{if}$. Note that

$$\mathcal{F}(k_{i\tau}) = \mathcal{F}(E_{ni} K_{ni\tau}) = E_{ni} \mathcal{F}(K_{ni\tau}) \tag{9}$$

due to the linearity of the Fourier operator $\mathcal{F}$, hence justifying the freedom of the contraction order. To optimize further, we would ideally compute $\mathcal{F}(E_{ni\tau} K_{ni\tau}) = \mathcal{F}(k_{i\tau})$ instead of $E_{ni} \mathcal{F}(K_{ni\tau})$, as the former requires performing the Fourier transform on a much smaller tensor (vector). Even in this simple example, we observe two important points: 1) the contraction order matters, 2) the placement of the Fourier operators matters.

To make things clearer, it is useful to think of the Fourier operator $\mathcal{F}_{ft}$ itself as an operand to be contracted. This will allow us to write the SSM operations as

$$y_{bit'} = \mathcal{F}^{-1}_{t'f} \big( \mathcal{F}_{ft} u_{bit} \big) \big( \mathcal{F}_{f\tau} E_{ni} K_{ni\tau} \big). \tag{10}$$

Here, we represented the Fourier operator as a matrix (i.e. the DFT matrix), but unlike standard matrix multiplication, we know that multiplying/contracting with a DFT matrix can be done via FFT, so that it will not incur the same computational complexity as the standard tensor contraction operation. Correctly accounting for the FFT complexity is important in determining the optimal order of contractions, which in this picture also includes the placement of the FFT operators.

This is a simple example where the optimal contraction is somewhat obvious and static, but there are cases where we have more terms to contract and the differences between the contraction paths are more subtle. And in these cases, the optimal contraction order is dynamically linked with the shapes of the tensor operands. Fortunately, there is a systematic way to evaluate the memory and compute requirement of an einsum contraction path (used prominently in packages such as `opt-einsum`) (Daniel et al., 2018), and we make a simple augmentation to this prescription to handle "contractions" involving FFT operators, while being somewhat mindful of the software and hardware backends (PyTorch and CUDA).

### 4.3 A PRACTICAL WALKTHROUGH: THE BOTTLENECK BLOCK

The main example that we will walk through here is the bottleneck layer example, or $\hat{y}_{bif} = \hat{u}_{bif} \overline{B}_{ni} E_{nmi} \hat{K}_{nmf} C_{jn}$, where 5 tensor operands are involved. If we include the Fourier operators as 3 additional operands, then we have a total of 8 operands, which yields a sufficiently complex design where the systematic optimization of contraction orders will show its power. See Listing 1 in Appendix C for a minimal pseudocode of the bottleneck block operations with the order of operations optimized, along with benchmarking on an A100 GPU.

In theory, it is possible to have `opt_einsum` handle optimizing all the contraction orders, and the pseudocode would appear much simpler (i.e. we only need one einsum expression in line 43 of Listing 1). However, there are certain SW and HW idiosyncrasies that make it more efficient to explicitly "force" certain contraction patterns. For instance: 1) complex tensors are not yet "natively" supported by CUDA, meaning that it is more efficient to operate on real tensors as much as possible. In other words, we perform computations in the complex frequency domain only if necessary; 2) `torch.compile` may not yet be able to identify kernel fusion opportunities within a sufficiently complex `torch.einsum` expression. Therefore, it may be more efficient to explicitly modularize and kernel-fuse certain contraction steps, particularly those that are memory-bound.

We try to achieve a happy medium between full automation with einsum expressions and full specialization with custom CUDA kernels. In other words, we "semi-manually" inspect all possible contraction paths and discard the clearly non-optimal ones. Then, we perform some amount of practical optimization for the "feasible" contraction paths, as we will explore in the following sections. Initially, for ease of exposition, we will continue our discussion treating the temporal domain

and frequency domain as almost equivalent, as the two can be easily traversed via FFTs (which is fairly light in compute). Only at the very end will we make the effort to separate the two domains as we begin to consider the optimal "insertion points" of the FFT operations, as the second-order optimization.

First, based on visual inspection of the bottleneck tensor network representation in Fig. 1, we can clearly see the first step should be to always "contract away" the inner edge $m$, or equivalently compute the kernel $k_{n\tau} = E_{nm}K_{nm\tau}$ first. However, we note that the basis kernels $K_{nm\tau}$ themselves are generated by $\Delta_n$ and $A_{nm}$. Therefore, under kernel fusion, it is possible to not even materialize the basis kernels $K_{nm\tau}$ in the GPU VRAM at all, which shaves off memory and computational requirements considerably during training. This is encapsulated in the `get_kernel` function in Listing 1, which is meant to be (jit-)compiled during training, for example with `torch.compile` or custom `triton` kernels. In the frequency domain, we are now left with the expression $\hat{y}_{bif} = \hat{u}_{bif}\overline{B}_{ni}\hat{k}_{nf}C_{jn}$, which still allows for many possible contraction paths in theory. However, heuristically speaking, we should try not to produce any intermediate tensor of dimension greater than 3, as it will be unfavorable to materialize and operate on it. Lemma 1 will heavily restrict the "feasible" contraction paths based on this criteria, a full proof of which is given in Appendix B.

**Lemma 1.** *Given the einsum expression $\hat{y}_{bif} = \hat{u}_{bif}\overline{B}_{ni}\hat{k}_{nf}C_{jn}$ arranged in this order, all intermediate tensors will have at most 3 dimensions, if and only if $\hat{u}$ (and intermediate tensors resulting from it) is contracted with only its neighboring operands.*

**Remark.** *An interpretation of Lemma 1 is that the contraction order should roughly follow the "natural order of operations" of the underlying state-space system, in some form of "associative" fashion. The proof sketch is to try contracting $u$ with its non-adjacent operands such as $k$ or $C$, and observe the appearance of high dimensional intermediate tensors, for instance, $\hat{u}_{bif}\hat{k}_{nf} = (\hat{u}\hat{k})_{binf}$*

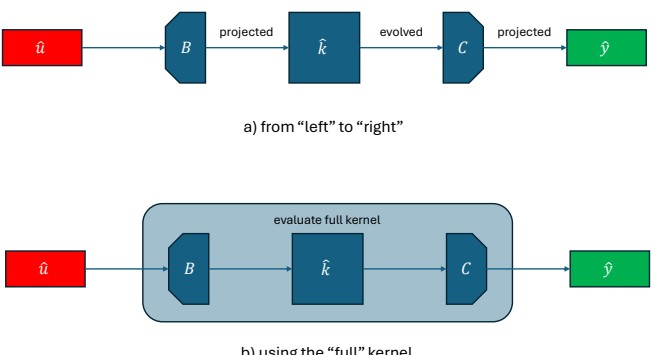

a) from "left" to "right"

b) using the "full" kernel

Figure 2: The two feasible patterns for performing the SSM bottleneck operations. a) Follow the "natural" order induced by the underlying SSM system: first project the inputs, then convolve them with the kernels, and finally project the outputs. b) Generate first the "full" SSM kernels, then convolve them with the inputs in one stage to get the output.

Under the restriction of Lemma 1, there are really only two "feasible" contraction patterns as showcased in Fig. 2. First, we can perform the operations from left to right, corresponding to projecting the input, performing the FFT convolution, and then projecting the output. This is just the "natural" order of operations induced by the underlying state-space system. Alternatively, we can build the "full kernel" first, then perform the full FFT convolution with the input as $\hat{x}_{bif}(\overline{B}_{ni}\hat{k}_{nf}C_{jn}) = \hat{x}_{bif}\hat{\mathbf{k}}_{jif} = \hat{y}_{bjf}$ in one step[3]. We see that the optimal contraction order is intimately linked with the dimensions of the tensor operands. More formally, the first "natural" order of contraction is only optimal when $\frac{1}{B} + \frac{1}{N} > \frac{1}{H} + \frac{1}{H'}$, as shown in Appendix B. For the first contraction pattern, if additionally the condition $N \leq H$ is met, then it is clear that we should

---

[3]This is how convolutional networks typically handle full convolutions, except there is no "kernel building" stage as the kernels are explicitly parameterized.

perform the input projection $x_{bnt} = u_{bit}B_{ni}$ first before performing the FFT on the projected input $\mathcal{F}(x_{bnt}) = \hat{x}_{bnf}$, as the projected input $x$ is smaller than the input $u$. Similarly, for the second contraction pattern, if additionally the condition $HH' \leq N$ is met, then we can produce the full kernel $\mathbf{k}_{ij\tau} = \overline{B}_{ni}k_{n\tau}C_{jn}$ first before performing the FFT on the full kernel $\mathcal{F}(\mathbf{k}_{ij\tau}) = \hat{\mathbf{k}}_{jif}$, as the full kernel $\mathbf{k}$ is actually smaller than $k$. These two contraction patterns are explicitly forced via the if statements in the `opt_fft_conv` function in Listing 1.

## 5 EXPERIMENTS

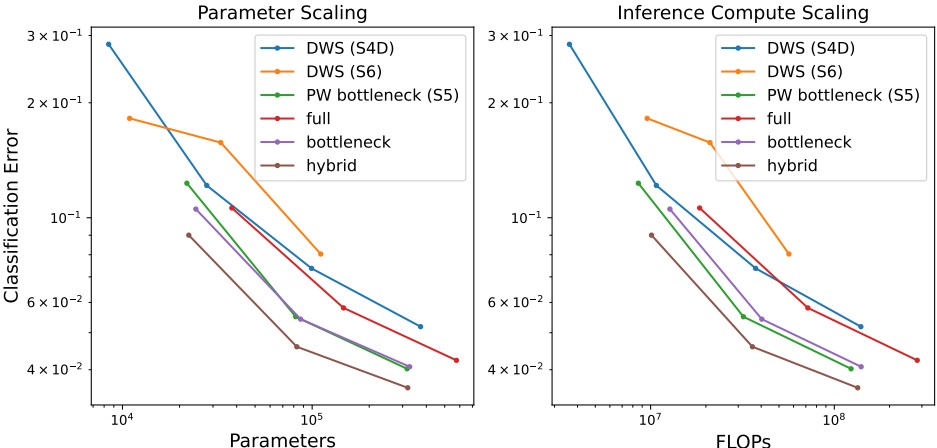

Figure 3: The scaling of the classification error versus the number of parameters and FLOPs per second during inference. The performance is evaluated on the SC35 testset. For each network variant in the ablation studies, we performed 10 training trials and took the median metric. DWS is short of depthwise-separable (with the S6 variant using selective scan), and PW is short of pointwise.

In our experiments, we study various configurations of Centaurus model architectures (e.g. classifier, hourglass, multi-streams) and block configurations (e.g. depthwise, bottleneck, full). This is to showcase the structural flexibility of our model and adaptation to different tasks, reminiscent of classical CNNs. At the same time, we try to keep consistent the training pipeline (e.g. optimizers and schedulers) and the auxiliary layers (e.g. normalization layers and activations), to isolate the effects of only the core architectural changes in the SSM blocks. See Appendix E for the training details and the basic design of the Centaurus block. We will mainly be comparing the following variants: 1) networks with all SSM layers being depthwise-separable (S4D-like and S6-like with data gating), 2) networks with all SSM layers being pointwise bottlenecks (S5-like), 3) networks with all SSM layers being bottlenecks, 4) networks with all SSM layers being full convolutions, and 5) hybrid networks with mixtures of SSM layer types (inspired by classical CNN designs). We will put a strong emphasis on the computational requirements of the network during *online inference*, in addition to the typical parameter count. This is complimentary to the theoretical and algorithmic components of our work, where optimal contractions are leveraged to speed up *training*. See Appendix D for a discussion of how the parameters and FLOPs are estimated for these blocks during online recurrence. We show that our hybrid Centaurus network can achieve competitive performance as a stand-alone SSM network. In addition, we show that it can be well augmented by inserting (explicit) convolutional layers or modules into it, following the modern trend of designing hybrid SSM models (Lieber et al., 2024; Ren et al., 2024; Patro & Agneeswaran, 2024; Glorioso et al., 2024). However, since we aim for solutions that can be configured for efficient online inference, we will avoid the use of any attention mechanism[4], though we expect it to also integrate well with Centaurus.

---

[4]Variants of the attention mechanism, such as sliding-window attention, can be used to improve the online inference efficiency. Any surrogate (linear) attention mechanism admitting a recurrent form (e.g. Linear transformer, RWKV, Mamba) can also be used. We think the Centaurus model can interface well with these blocks, but do not consider them here to limit the scope of this work.

## 5.1 KEYWORD SPOTTING

We begin with the simple task of keyword spotting (KWS) with raw waveforms, using the Google Speech Commands 35-class subset (Warden, 2018), or SC35. For each SSM configuration, we train networks of various sizes, to estimate the scalability of the performance with respect to the number of parameters and FLOPs per second during online inference. See Fig. 3 for the scalability plots, where our hybrid architecture outperformed all the homogeneous counterparts. All the network variants we tested have six layers. Our Centaurus hybrid model contains two layers of full SSM blocks, followed by two layers of bottleneck blocks, followed by two final layers of pointwise bottleneck blocks (S5). This is in accordance with the classical CNN design philosophy of using sparser connective structures at deeper layers where the number of channels becomes large. The training and network details are given in Appendix E.1, along with a comparison against other deep SSMs, where our hybrid network achieved similar performance with roughly 100 times fewer FLOPs. Additionally in Appendix E.1, we compare against a hybrid variant where the projection matrices ($\overline{B}$, $C$, and $E$) are complex (similar to the original S4D and S5 layers), but it did not outperform the real counterpart; furthermore, we compare against variants with S6 layers of different state dimensions.

## 5.2 SPEECH ENHANCEMENT

Table 1: Comparing real-time audio denoising networks. Note that only the Centaurus network requires no additional signal processing (e.g. STFT). To boost the denoising performance, we can prepend every SSM block with a causal depthwise Conv1d layer with a kernel size of 4 (last row).

| Model | PESQ (VB-DMD) ↑ | Parameters | FLOPs / sec |
|---|---|---|---|
| **Real-time Baselines** (with processing) | | | |
| FRCRN (Zhao et al., 2022) | 3.21 | 6.9M | 76.2G |
| DeepFilterNet3 (Schröter et al., 2023) | 3.16 | 2.13M | 0.688G |
| SEMamba (causal) (Chao et al., 2024) | 2.76 | 3.60M | 0.76G |
| PercepNet (Valin et al., 2020) | 2.73 | 8.00M | 1.60G |
| DEMUCS (Defossez et al., 2020) | 2.65 | 33.53M | 15.44G |
| Centaurus (DWS) | - | 0.71M | 0.39G |
| Centaurus (DWS data-gated, or S6) | - | 0.66M | 0.50G |
| Centaurus (bottleneck) | - | 0.87M | 1.15G |
| Centaurus (PW bottleneck) | 3.06 | 0.83M | 0.65G |
| Centaurus (full) | 3.04 | 2.53M | 1.12G |
| **Centaurus** (hybrid) | 3.12 | **0.51M** | **0.29G** |
| **Centaurus** (hybrid, with Causal Conv) | **3.25** | **0.51M** | **0.29G** |

Here, we focus on the task of real-time speech denoising directly on raw audio waveforms. The macro network architecture used in this experiment is adapted from a deep SSM hourglass autoencoder proposed in Pei et al. (2024), where the SSM blocks were originally configured as DWS blocks (S4D). Here, we modify the network to include multiple variants of SSM blocks to enhance its connective flexibility, and in addition perform real-time denoising on raw waveforms directly in the $[-1, +1]$ range without one-hot encoding (Goel et al., 2022) or spectral processing (Schröter et al., 2023). We will test variants of this network by simply swapping out the pointwise-bottleneck SSM blocks with other variants. The encoder architecture is based on the 6-layer KWS network backbone (see Section 5.1), and the decoder is a reflection of the encoder. The performance of the network is evaluated on the VoiceBank + DEMAND (VB-DMD) testset and given in Table. 1. Details of the network architectures and the training pipeline are given in Appendix E.2.

Surprisingly, the DWS and bottleneck homogeneous variants suffered from severe training plateaus, and were not able to generate any meaningful audio samples. In addition, using the data-gating mechanism (S6 layers) did not solve the issue. The variant with homogeneous full SSM blocks did manage to train, but despite its size, it did not achieve the best performance. Out of all variants, the hybrid Centaurus model achieved competitive PESQ scores, while requiring much fewer parameters and FLOPs. There are concurrent works applying deep SSMs to speech enhancement (in cases

outperforming Centaurus), but they configure the SSM layers as bidirectional, which sacrifices the online capability of the network (Zhang et al., 2024; Chao et al., 2024).

## 5.3 Towards end-to-end speech recognition with SSMs

Table 2: The reported metrics are word error rates (WERs) for the clean/other splits, on the Librispeech test/dev-sets. The FLOPs are estimated based on a 20.5-second audio segment.

| Model | test ↓ | dev ↓ | Parameters (M) | FLOPs (G) |
|---|---|---|---|---|
| **Offline (full-context)** | | | | |
| Full Conv (Zeghidour et al., 2018) | 3.3 / 10.5 | 3.1 / 9.9 | | |
| Transformer (Zhang et al., 2020) | 3.1 / 7.3 | | 29 | |
| ContextNet (Han et al., 2020) | 2.4 / 5.4 | | 31.4 | |
| Wav2Vec2 (Baevski et al., 2020) | 2.1 / 4.8 | 1.8 / 4.7 | 94.4 | 285 |
| Conformer (Gulati et al., 2020) | 2.0 / 4.3 | | 30.7 | 45.2 |
| **Online (streaming)** | | | | |
| Transformer | 5.0 / 11.6 | | 18.9 | |
| ContextNet | 4.5 / 10.0 | | 31.4 | |
| Conformer | 4.6 / 9.9 | | 30.7 | |
| **Centaurus (online, no attention)** | | | | |
| Base (full SSM) | 6.0 / 13.1 | 5.9 / 13.1 | 12.4 | 20.6 |
| with FFN | 5.4 / 11.5 | 5.2 / 11.3 | 18.0 | 32.1 |
| with causal conv-block | 4.8 / 10.6 | 4.8 / 10.5 | 23.6 | 43.7 |
| with Mamba macro-block | 4.4 / 10.2 | 4.3 / 9.9 | 29.9 | 46.9 |

Since this is not a work focused on training methodology, we opt for a rather simple supervised/distillation pipeline here[5]. At a high level, we use a hybrid SSM backbone similar to our KWS hybrid network (see Section 5.1), but wider and deeper (see Appendix E.3). We do not use any additional lexicon or language model decoding (Zhang et al., 2020; Gulati et al., 2020). And unlike previous works, we are not integrating SSM layers as auxiliary layers in off-the-shelf transformer models (Miyazaki et al., 2023; Shan et al., 2023). This means that the Centaurus network is fully SSM based end-to-end, without any non-linear recurrent mechanism (self-conditioned decoding) or attention mechanism. In Table. 2, we report the performance and size of the base Centaurus model on Librispeech (Panayotov et al., 2015), along with two augmented variants: 1) one with a feed-forward (FF) module appended to every SSM block, 2) one with a convolutional (Conv) module appended, 3) and one with the Mamba gated macro-architecture (Gu & Dao, 2023) wrapped around our core SSM layers. We describe the architecture of these modules in Appendix E.3. Being a fully online-inference ASR model with minimal optimization, the Centaurus network still remains competitive with "benchmark" ASR networks like Conformer and Wav2Vec2 in streaming mode.

## 6 Conclusion

We introduced Centaurus, composed of SSM blocks with flexible connectivity structures, trained using optimized tensor contractions. We designed task-specific realizations of the Centaurus network balancing between size and performance. As inspired by classical CNN designs, we used blocks with dense connections in the shallower layers and opted for sparser connections in the deeper layers. The hybrid networks achieved competitive results on multiple audio processing tasks, while being fully SSM based. In the future, it may be fruitful to explore how the data-gating mechanism can be applied to the Centaurus network, and explore its scalability to tasks that are of larger scales, such as language modeling. In addition, it may also be of interest to explore SSM convolutional structures in higher dimensions, incorporating additional structures such as stride, dilation, and transposition, to see whether Centaurus can be effective for computer vision tasks as well.

---

[5]To train an ASR model from scratch, the prevailing method is typically to perform self-supervised learning on a large amount of unlabeled speech, then to fine-tune on a relatively small amount of labeled speech.

## ACKNOWLEDGMENT

We would like to thank Nolan Ardolino, Olivier Coenen, M. Anthony Lewis, Sidharth (in alphabetical order) for many inspiring discussions. In addition, we thank Dhvani Kothari and Kurt Manninen for producing spectacular demos using the Centaurus network. We also thank Daniel Keller for providing a critical proofreading of the manuscript. Finally, we thank the ICLR reviewers for offering much helpful feedback for making our work more complete.

## ETHICS STATEMENT

We presented a novel state-space network for several important audio processing tasks. We made sure that our networks are lightweight for both training and inference, so that they can be easily experimented with by both researchers and hobbyists alike. This can be done without requiring too much computational resource, as part of our commitment to democratizing AI and reducing carbon emissions. All of the datasets used in our work are publicly available, with no private or sensitive data used in our experiments.

## REPRODUCIBILITY STATEMENT

We gave the full details of the training pipeline of our networks in Section 5 of the main text and Appendix E. Listing 1 in Appendix C provides a pseudocode that can be easily adapted into PyTorch code, and we additionally made available the model source code at github.com/Brainchip-Inc/Centaurus.

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

## A    CANONICAL FORM OF A STATE-SPACE MODEL

It is a generic property (though not always true) that a diagonal form exists for the state-space model, meaning that we can almost always assume $\overline{A}$ to be diagonal, at the expense of potentially requiring $\overline{B}$ and $C$ to be complex matrices. Since the original system is a real system, the diagonal $\overline{A}$ matrix can only contain real elements and/or complex elements in conjugate pairs. In this work, we sacrifice a slight loss in expressivity by continuing to restrict $\overline{B}$ and $C$ to be real matrices[6] and letting $\overline{A}$ be a diagonal matrix with all complex elements, but not restricting them to come in conjugate pairs. During online inference, we then need to maintain the internal states $x$ as complex values but only need to propagate their real parts to the next layer.

Recall that the discretized state-space system is given by

$$x[t+1] = \overline{A}x[t] + \overline{B}u[t], \qquad y[t+1] = Cx[t+1] + Du[t+1], \tag{11}$$

To see that the term $Du$ can be absorbed into the state if we double the internal states, we simply need to make the following substitution

$$\overline{B} \rightarrow \left[ \frac{\overline{B}}{\mathbf{1}_{N \times H}} \right] \quad \overline{A} \rightarrow \left[ \begin{array}{c|c} \overline{A} & \mathbf{0}_{N \times N} \\ \hline \mathbf{0}_{N \times N} & \mathbf{0}_{N \times N} \end{array} \right], \quad C \rightarrow \left[ \begin{array}{c|c} C & D \end{array} \right]. \tag{12}$$

In other words, we can trivially set aside a set of "memoryless" internal states whose sole purpose is to duplicate the input $u$, effectively routing directly the inputs to the outputs.

If we allow $\overline{B}$ and $C$ to be complex matrices, then we can further assume $\overline{A}$ to be diagonal without any loss of generality. To see why, we simply let $\overline{A} = P^{-1}\Lambda_A P$ (where $\Lambda_A$ is the diagonalized $\overline{A}$ matrix, and $P$ is the similarity matrix) and observe the following:

$$
\begin{aligned}
\forall t, \qquad k[t] &= C(P^{-1}\Lambda_A P)^{t-1}\overline{B} \\
&= CP^{-1} \underbrace{\Lambda_A(PP^{-1})...\Lambda_A(PP^{-1})}_{\text{repeat } t-1 \text{ times}} P\overline{B} \\
&= (CP^{-1})\Lambda_A^{t-1}(P\overline{B}) \\
&= C'\Lambda_A^{t-1}\overline{B}',
\end{aligned}
\tag{13}
$$

where $\overline{B}'$ and $C'$ are complex matrices that have "absorbed" the similarity matrix $P$, but WLOG we can just redefine them to be $\overline{B}$ and $C$. Since $\overline{A}$ is a real matrix, the complex eigenvalues in $\Lambda_A$ must come in conjugate pairs. And WLOG we can again redefine $\Lambda_A$ as $\overline{A}$. Note that in this section, we

---

[6]This is in deviation of prior works, such as S4D and S5, where $\overline{B}$ and $C$ are complex.

only need to ensure that $\overline{A}$ is a square matrix, and all the other state-space matrices and state vectors can be of arbitrary shape, as long as they are conformable.

Since we still want to work with real features[7], we then only take the real part of the impulse response kernel as such:

$$k[\tau] = \Re(C\overline{A}^\tau \overline{B}) = C\Re(\overline{A}^\tau)\overline{B}, \tag{14}$$

which equivalently in the state-space equation can be achieved by simply letting $y[t] = C\Re(x[t])$. This means that during online inference, we need to maintain the internal states $x$ as complex values, but only need to propagate their real parts to the next layer. This restriction of propagating real features is not required, and is merely for better CUDA support, as discussed in Section 4.3.

We can then define the basis kernels to be $K_n[\tau] = \Re(\overline{A}^\tau)$. Furthermore, we use the hat symbol to denote the Fourier transform of the features, or $\hat{x} = \mathcal{F}(x)$ where $\mathcal{F}$ is the Fourier operator. To be more explicitly, if $\tau \in \{0, 1, 2, ..., T-1\}$, then we have

$$\hat{K}_{nf} = \mathcal{F}(K_n[\tau]) = \sum_{\tau=0}^{T-1} \frac{1}{\sqrt{T}} K_n[\tau] \exp\left(-\frac{2\pi i \tau f}{T}\right), \tag{15}$$

noting that the index $n$ is not of importance, and can be replaced with any (multi)index when needed. For now, we ignore the artifacts of cyclic convolution (needing padding to ensure causality) and algorithmic optimizations[8], except for the obvious fact that the Fourier transform can be computed efficiently via the Fast Fourier Transform (FFT) algorithm (Cooley & Tukey, 1965). At this stage, we then have a condensed `einsum` expression in the frequency domain that allows us to easily see the interactions between the input signals, basis kernels, and system matrices.

## B  BOTTLENECK SSM CONTRACTION ORDERS

Recall that Lemma. 1 states that: Given the einsum expression $\hat{y}_{bif} = \hat{u}_{bif}\overline{B}_{ni}\hat{k}_{nf}C_{jn}$ arranged in this order, all intermediate tensors will have at most 3 dimensions, if and only if $\hat{u}$ (and intermediate tensors resulting from it) is contracted with only its neighboring operands.

*Proof.* If we contract $\hat{u}_{bif}$ with $\hat{k}_{nf}$, this immediately results in a 4D tensor $(\hat{u}\hat{k})_{binf}$. Similarly, if we contract $\hat{u}_{bif}$ with $C_{jn}$, this immediately results in a 5D tensor $(\hat{u}C)_{bjinf}$. This means that we can only contract $\hat{u}_{bif}$ with $\overline{B}_{ni}$ first (its only neighboring operand), resulting in the 3D tensor $(\hat{u}B)_{bnf}$.

At the second stage, if we contract $(\hat{u}\overline{B})_{bnf}$ with $C_{jn}$, this will result in a 4D tensor $(\hat{u}\overline{B}C)_{bjnf}$. This means that we can only contract $(\hat{u}\overline{B})$ with $\hat{k}_{nf}$ (again its only neighboring operand), resulting in the 3D tensor $(\hat{u}\overline{B}\hat{k})_{bnf}$.

For the remaining contraction paths, we have to verify that any contraction paths within $\overline{B}_{ni}\hat{k}_{nf}C_{jn}$ (not including $\hat{u}$) will only result in intermediate tensors of dimension at most 3: $\overline{B}_{ni}\hat{k}_{nf} = (\overline{B}\hat{k})_{nif}$, $\hat{k}_{nf}C_{jn} = (\hat{k}C)_{jnf}$, and $\overline{B}_{ni}C_{jn} = (\overline{B}C)_{jni}$. $\square$

An interpretation of Lemma 1 is that the contraction order should roughly follow the "natural order of operations" of the underlying state-space system, in some form of "associative" fashion. For example, it clearly makes little sense to contract the input $u$ directly with the output project matrix $C$ before even passing the input through the internal states first, and this is formally reflected as the production of a 5D intermediate tensor. Note that this is not to say that the contractions of an einsum expression should be restricted to neighboring operands. In fact, einsum expressions are agnostic to the ordering of operands, meaning that contractions can be performed on any two operands at any stage. The discussion here is only specific to the state-space system $\hat{y}_{bif} = \hat{u}_{bif}\overline{B}_{ni}\hat{k}_{nf}C_{jn}$, for

---

[7]This is *a prior* not required, as technically we can configure our network as complex-valued to handle complex features. However, we do not explore this configuration in this work (complex-valued neural networks).

[8]As an example, for a real signal, we only need to extract half the Fourier modes, or up to the Nyquist frequency.

which this artificial ordering restriction is only needed for efficiency concerns, and not functional correctness (which will be guaranteed regardless of the contraction path taken).

Under the restriction of Lemma 1, there are really only two "feasible" contraction patterns. First, we can perform the operations from left to right, corresponding to projecting the input, performing the FFT convolution, and then projecting the output. This is just the "natural" order of operations induced by the underlying state-space system. Alternatively, we can build the "full kernel" first, then perform the full FFT convolution with the input as $\hat{x}_{bif}(\overline{B}_{ni}\hat{k}_{nf}C_{jn}) = \hat{x}_{bif}\hat{\mathbf{k}}_{jif} = \hat{y}_{bjf}$ in one step[9]. If we only focus on the computational requirements of the forward pass[10], then the first contraction order will result in $BNHF + BNF + BH'NF \approx BNF(H + H')$ units of computation. The second contraction order will result in $H'NH + JNHF + BH'HF \approx HH'F(B + N)$ units of computation. Therefore, we see that the optimal contraction order is intimately linked with the dimensions of the tensor operands. More formally, the first "natural" order of contraction is only optimal when $BNF(H + H') < H'HF(B + N)$ or $\frac{1}{B} + \frac{1}{N} > \frac{1}{H} + \frac{1}{H'}$.

## C  BOTTLENECK SSM BLOCK PSEUDOCODE AND BENCHMARKING

Listing 1: Contraction operations for the bottleneck SSM block

```
1  # assumes opt_einsum backend is enabled
2
3  def fft_conv(equation, x, k, *args):
4      # FFT conv with additional non-sequence tensors as args
5      L = x.shape[-1]
6      x_f = rfft(x, length=2*L)
7      k_f = rfft(k, length=2*L)
8      y_f = einsum(equation, x_f, k_f, *args)
9      return irfft(y_f)[..., :L]
10
11 @compile
12 def get_kernel(delta, A, E, length):
13     dtA = einsum('n,nm->nm', delta, A)
14     K = einsum('nm,t->nmt', dtA, arange).exp()
15     return einsum('nmt,nm->nt', K.real, E)
16
17 def opt_fft_conv(u, k_real, B_discrete, C):
18     # all the input arguments are real tensors
19     # u: (batch, H_in, L)
20     # k_real: (N, L)
21     # B_discrete: (N, H_in)
22     # C: (H_out, N)
23
24     # force certain contraction paths based on the tensor shapes
25     batch, H_in, _ = u.shape
26     H_out, N = C.shape
27
28     # project inputs, perform FFT conv, then project outputs
29     if (1 / H_in + 1 / H_out) < (1 / batch + 1 / N):
30         if N <= H_in:
31             x = einsum('bit,ni->bnt', u, B_discrete)
32             x = fft_conv('bnt,nt->bnt', x, k_real)
33             return einsum('bnt,jn->bjt', x, C)
34     # evaluate the full kernel, then perform full FFT conv with inputs
35     # directly in one stage
36     else:
37         if H_in * H_out <= N:
38             k_full = einsum('jn,ni,nt->jit', C, B_discrete, k_real)
39             return fft_conv('bit,jit->bjt', x, k_full)
```

---

[9]This is how convolutional networks typically handle full convolutions, except there is no "kernel building" stage as the kernels are explicitly parameterized.

[10]Under reverse-mode automatic differentiation, it is not hard to show that the compute units required for the backward pass are 2 times that of the forward pass, for any einsum expression.

```
40
41      # can simply just return this and ignore the above
42      # if some loss of efficiency can be accepted
43      return fft_conv('bit,nt,ni,jn->bjt', u, k_real, B_discrete, C)
44
45  def ssm_bottleneck(u, delta, A, B, C, E):
46      length = u.shape[-1]
47      k_real = get_kernel(delta, A, E, length)
48      B_discrete = einsum('n,ni->ni', delta, B)   # ZOH discretization
49      return opt_fft_conv(u, k_real, B_discrete, C)
```

Here, we acknowledge that this semi-manual method, though efficient, is not the most elegant or scalable approach, as it requires some amount of "hard coding" for every SMM block variant. Fortunately, the bottleneck block we walked through in this section is the most involved example, and the other SSM block variants explored in this work are much simpler in terms of feasible contraction paths. Nevertheless, we are working on an extension of the opt_einsum library specifically for general temporal sequence modeling using tensor networks that can: 1) identify any kernel-fusion opportunities in the underlying tensor network, 2) identify the optimal "insertion points" for FFT and iFFT operations accounting for the preference of real tensors over complex ones.

We compare the training performance of the contraction-optimized SSM block as given in Listing 1 against the naive contraction order following the "natural" order of input projection, state evolution, and output projection (as typically done for other SSM networks). To isolate just the effect of optimal contraction, we focus on the total time it takes for the forward plus backward pass on a single bottleneck SSM block. We perform the benchmark in fp32 precision on $1\times$ A100 40GB SXM4 with PyTorch 2.5.1 under CUDA 12.4. The baseline dimensions are batch $= 256$, $H = 16$, $H' = 32$, length $= 2048$, $N = 256$, and $M = 16$. We perform three separate scaling studies along the batch, $N$, and length dimensions respectively, and scale them from 32 to 2048.

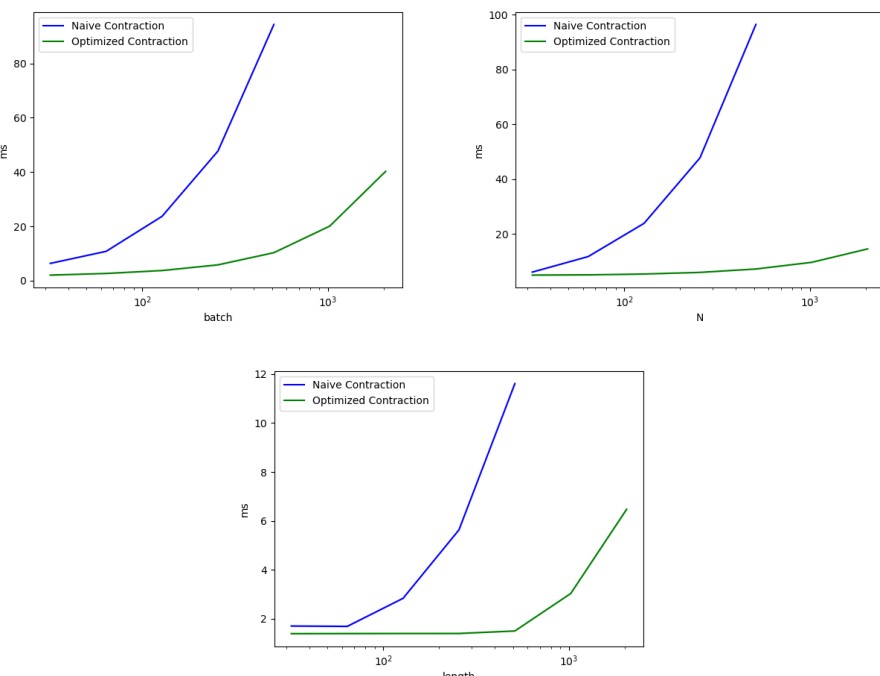

Figure 4: The training time (forward plus backward pass time) scaling with respect to the batch, state, and length dimensions. Note that the x-axis is in log scale.

## D   ESTIMATION OF PARAMETERS AND FLOPS

It is very tempting to train network variants that are feasible by optimal contractions, but eventually turn out to be incredibly memory or computationally expensive during inference. An example would be to aggressively use "full" SSM blocks with large channels and internal states, which will result in the internal states of size (out channels, input channels, states) needing to be maintained and updated during inference. Even though the memory and computational bottlenecks can be mitigated during training via optimal contractions, this is not possible during online inference time if none of the internal states can be "contracted away"[11]. Therefore, following classical designs of lightweight convolutional networks (Howard et al., 2017; Sandler et al., 2018), we use full SSM blocks sparingly only in the first few layers where the channel dimensions are still small, and we will mostly be using (pointwise) bottleneck blocks in the deeper layers where the channel dimensions become larger.

Note that the number of parameters and FLOPs differ between online inference and training. For inference parameters, certain learnable parameters can be absorbed into the system matrices, such as the $\Delta$ parameter. For inference computations, the number of FLOPs is always linear with respect to the sequence length; however, this is often outweighed by the suboptimal order of operations imposed by the explicit materialization of internal states. In Table. 3, we provide the parameter count and the FLOPs per recurrent step when various SSM block variants are configured for online inference. There are a couple of points to note:

- During online inference, the internal states need to be explicitly maintained and updated, meaning that the order of operation is always first the input projection, internal states update, output projection, and an optional mixer layer (for S4D).

- The internal states are maintained as complex tensors, but we only take the real parts for the output projection. Note that a complex weight is doubled the parameter count of the real counterpart, and a complex multiplication is 6 times the FLOPs of the real counterpart (4 multiplications + 2 additions). In addition, updating the internal state with the projected real input requires 1 FLOP, and 2 FLOPs if the project input is also complex.

- In the case where the projection matrices (e.g. $B$ and $C$) are also complex, a complex dot product (used in matrix multiplications) is performed and will incur additionally 2 extra FLOPs per accumulation, resulting in a total of 8 FLOPs per matrix element. However, if we only need the real part of the projected outputs (or analogously only having real inputs), the projection operation will incur half the number of FLOPs compared to the full complex projection, or 4 FLOPs per matrix element.

- We do not count peripheral parameters and FLOPs for biases or affine transformations in normalization layers, as they are negligible compared to the SSM operations.

In Centaurus, we restrict all model parameters except for the state transition matrix $\overline{A}$ to be real. If we make the projection matrices complex, we can then recover the original S4D and S5 implementations as DWS and PW bottleneck blocks respectively. Following the original implementations, we still restrict the inputs and outputs of the SSM layers to be real. Besides using complex projection matrices, there are some additional idiosyncratic differences:

- The original S4D layer uses a standard where a complex internal state is considered to be two states, whereas S5 and Centaurus do not and simply consider it as one full state.

- The original S4D layer uses the GLU activation, meaning that the pre-activations hence the $C$ matrix will have double the size compared to Centaurus.

Note that if a direct residual connection is added to the SSM block, there will only be a trivial amount of FLOPs added equaling $H'$. In the case where the residual connection is a projection operation (necessary if $H \neq H'$), then there will be $HH'$ additional parameters added and roughly $2HH'$ additional FLOPs. Residual projections will be used in our Centaurus networks for keyword spotting and automatic speech recognition.

---

[11]Of course, this is a non-problem if the network is to perform inference in an offline setting, where no internal states need to be explicitly materialized during inference, and the flexibility of contraction order is restored.

Table 3: The number of parameters and FLOPs per recurrent step during online inference mode for the SSM block variants. We assume that all the optimizations for online inference have already been done, including the pre-computation of the SSM discretization and diagonalization, except for the S6 block whose SSM matrices need to be dynamically generated.

| SSM Block | Parameters | FLOPs / step |
|---|---|---|
| Depthwise | $3HN$ | $9HN$ |
| Depthwise Separable (S4D-like) | $3HN + HH'$ | $9HN + 2HH'$ |
| Pointwise Bottleneck (S5-like) | $HN + 2N + H'N$ | $2HN + 7N + 2H'N$ |
| Bottleneck | $HN + 3NM + H'N$ | $2HN + 9NM + 2H'N$ |
| Full | $3HH'N$ | $9HH'N$ |
| Depthwise (complex) | $4HN$ | $11HN$ |
| Depthwise Separable (complex) | $4HN + HH'$ | $11HN + 2HH'$ |
| Pointwise Bottleneck (complex) | $2HN + 2N + 2H'N$ | $4HN + 8N + 4H'N$ |
| Bottleneck (complex) | $2HN + 4NM + 2H'N$ | $4HN + 16NM + 4H'N$ |
| Full (complex) | $4HH'N$ | $11HH'N$ |
| Depthwise Separable (S6-like) | $4HN + H^2/r + HH'$ | $14HN + 4H^2/r + 2HH'$ |
| Mamba | $8H^2 + 8HN + 4H^2/r$ | $16H^2 + 28HN + 16H^2/r$ |

For the S6 layer used in Mamba (Gu & Dao, 2023), we only extract the core selective scan operation, and replace the depthwise SSM layer with it for our ablations, while still maintaining the output mixer layer. Note that every parameter and operation involved in the S6 layer is fully in real space. The core of the S6 layer is the selective scan mechanism, where the state matrices are data-controlled. This mechanism is only meaningful if the channel (feature) dimension is greater than 1, so for mono-channel layers we will always fallback to the standard depthwise layer. The estimation of parameters and FLOPs for this layer is as follows:

- For the data-gating operation, the generation of $\Delta$ requires roughly $H^2/r$ parameters and $4H^2/r$ FLOPs, where $r$ is the low-rank factor and the low-rank dimension being $\lceil H/16 \rceil$, chosen to be 16. The generation of the $B$ and $C$ matrices requires a total of $2HN$ parameters and $4HN$ FLOPs.

- The discretization process has to occur dynamically due to the dynamicity of the $\Delta$ parameter. It does not incur additional parameters but does require additional FLOPs. This involves applying softplus to $\Delta$, performing an element-wise multiplication with the input and the transition matrix $A$, and exponentiating the latter. We conservatively estimate the FLOPs required to be $5HN$, under the assumption that the `expf` operation takes at least 4 FLOPs per element.

- Finally, we have the actual depthwise SSM operations, which in real space, incur $2HN$ parameters and $5HN$ FLOPs.

In the original Mamba block, $r = 16$ is kept constant. The channel dimension $H$ also gets doubled before passing into the core S6 layer. On top of this, the network has an additional non-linear "gating" path that also has $2H$ features. The input and output projection layers allowing for this macro structure will then result in a total of $8H^2$ parameters and $16H^2$ FLOPs, in addition to the core S6 operations. There is also a lightweight causal depthwise Conv1D layer that has negligible parameters and FLOPs. The Mamba block is like a "bottleneck" block that does not alter the channel dimension $H$, hence we can perform a channel projection during downsampling to project $H$ to $H'$ (see next Section).

# E  EXPERIMENT DETAILS

Unless otherwise mentioned, all of our network variants are trained with:

- AdamW optimizer with the PyTorch default configs

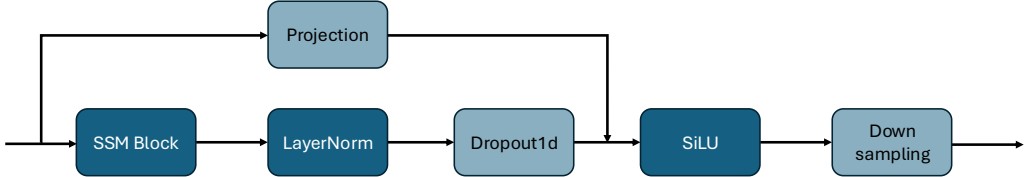

Figure 5: We use a basic design for our Centaurus block, where the lighter-shaded blocks are optional. The skip path is typically just an identity connection, but can be a pointwise projection layer to change the channel dimension. Similarly, the (optional) downsampling layer can be a simple average pooling to preserve the channel dimension, or a downsampling convolutional layer to project the channel dimension at the same time. The downsampling layer can be replaced by the upsampling layer if needed (e.g. in a decoder block of an auto-encoder).

- a cosine decay scheduler with a linear warmup period equal to 0.01 of the total training steps, updating after every optimizer step
- gradient clip value of 1
- layer normalization (over the feature dimension) with elementwise affine parameters
- SiLU activation
- no dropout layers except for the keyword-spotting network

Additionally, we trained with automatic mixed precision (AMP) along with `torch.compile`, except for the FFT convolution operations which are performed in full fp32 precision and without compilation (due to inability to handle complex data types currently).

Training configs not mentioned in the above list will be mentioned separately in the following individual subsections. We initialize the $B$ and $C$ projection matrices with the standard Kaiming uniform distribution. For the initialization of the $\Delta$ and $A$ parameters, we follow the S4D initialization strategy due to its simplicity. In other words, we set $\Delta$ to be geometrically ranged from 0.001 to 0.1, and $A_n = -1/2 + in\pi$. We suspect any sensible initialization method should also work fine. Below, we explicitly mention the dimension for which the initialization range is applied, with the indices denoting the dimensions of the $\Delta$ and $A$ parameters (slight abuse of notation).

- For S4D, we initialize $\Delta_{cn}$ over the $c$ dimension, and $A_{cn}$ over the $n$ dimension.
- For S5, we split the internal states into groups of 4. $\Delta_n$ is initialized across the groups, and $A_n$ is initialized within each group.
- For the full SSM block, we initialize $\Delta_{in}$ over the $i$ dimension, and $A_{jin}$ over the $n$ dimension.
- For the bottleneck block, we initialize $\Delta_n$ over the $n$ dimension, and $A_{nm}$ over the $m$ dimension.

All of our training runs and trials are done with PyTorch with `torch.compile` enabled except for operations involving complex numbers (e.g. FFTs). In addition, we enabled `tensorfloat32` for matrix multiplications, and the `opt_einsum` backend for all `torch.einsum` operations.

### E.1 KEYWORD SPOTTING

For all our trials in this experiment, we

- train for 200 epochs
- use a learning rate of 0.01 with a weight decay 0.05
- use a linear warmup period of 0.1 for the scheduler.
- Dropout1d with probability 0.1, only applied if the number of features is greater than 4

| Architecture | Channels | States |
|---|---|---|
| depthwise-separable (S4D/S6-like) | $[2, 4, 8, 16, 32, 64]$ | 4 |
| | $[4, 8, 16, 32, 64, 128]$ | 4 |
| | $[8, 16, 32, 64, 128, 256]$ | 4 |
| | $[16, 32, 64, 128, 256, 512]$ | 4 |
| pointwise bottleneck (S5-like) | $[2, 4, 8, 16, 32, 64]$ | $[4, 8, 16, 32, 64, 128]$ |
| | $[4, 8, 16, 32, 64, 128]$ | $[8, 16, 32, 64, 128, 256]$ |
| | $[8, 16, 32, 64, 128, 256]$ | $[16, 32, 64, 128, 256, 512]$ |
| full | $[2, 4, 8, 16, 32, 64]$ | 4 |
| | $[4, 8, 16, 32, 64, 128]$ | 4 |
| | $[8, 16, 32, 64, 128, 256]$ | 4 |
| bottleneck | $[2, 4, 8, 16, 32, 64]$ | $[4, 8, 16, 32, 64, 128]$ |
| | $[4, 8, 16, 32, 64, 128]$ | $[8, 16, 32, 64, 128, 256]$ |
| | $[8, 16, 32, 64, 128, 256]$ | $[16, 32, 64, 128, 256, 512]$ |
| full $\times 2$ + bottleneck $\times 2$ + S5 $\times 2$ | $[2, 4, 8, 16, 32, 64]$ | $[4, 4, 16, 32, 64, 128]$ |
| | $[4, 8, 16, 32, 64, 128]$ | $[4, 4, 32, 64, 128, 256]$ |
| | $[8, 16, 32, 64, 128, 256]$ | $[4, 4, 64, 128, 256, 512]$ |

Table 4: The different network variants tested. For the bottleneck blocks, it is always assumed that the number of sub-states is 4.

- perform training on a single NVIDIA A30 GPU with a batch size of 512

The network contains six SSM blocks, with output channels and internal states given in Table. E.1. Besides the first block of the network, every SSM block is a residual block where the skip path is a pointwise projection. We apply LayerNorm after the SSM operations but before the skip merge, and apply the SiLU activation after the skip merge, following the classical ResNet residual block design. After each SSM layer, we perform 1D average pooling with window sizes $\{4, 4, 2, 2, 2, 2\}$ respectively. We apply this network over one-second durations of raw audio waveforms, and a global-average-pooling (GAP) on the features generated by the SSM backbone, followed by a 2-layer MLP classification head[12].

In addition, we test against a hybrid network variant where the projection matrices are made complex, shown in Fig. 6. Over the model sizes tested, it appears that real variants outperform the complex variants when accounting for the model parameters and inference FLOPs. Furthermore, we compare against network variants with S6 layers having different state dimensions, as the number of states appears to be an important factor for data-controlled SSM layers (Gu & Dao, 2023).

To compare with the top results achieved on KWS by other state-space models, we use our largest Centaurus hybrid network architectured as given in Table. E.1. We report the performance and size of each model in Table. 5 on the Speech Command 10-class subset (SC10), as it is a common benchmark for all the models. Centaurus achieves SOTA results with less parameters and orders of magnitude less FLOPs.

### E.2 RAW SPEECH DENOISING

The high-level training pipeline for the raw audio denoising model is to simply generate synthetic noisy audios by randomly mixing clean and noise audio sources. The noisy sample is then used as input to the model, and the clean sample is used as the target output. For the clean training samples, we use the processed VCTK and LibriVox datasets that can be downloaded from the Microsoft DNS4 challenge. We also use the noise training samples from the DNS4 challenge as well, which contains

---

[12]The application of the GAP layer is in accordance with previous works. In a practical real-time inference setting, the GAP layer will likely be removed, and the classification head will be likely attached directly to the streaming output features of the SSM backbone. Standard post-processing techniques such as majority-filtering and early prediction can then be applied.

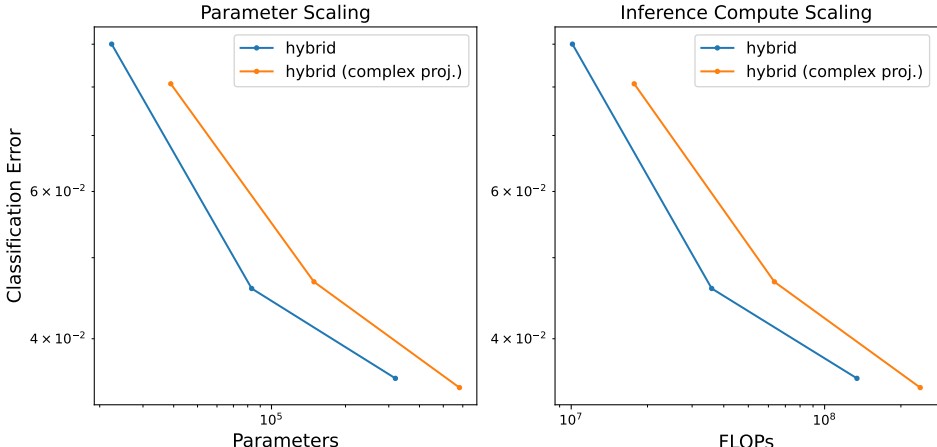

Figure 6: The scaling of the multiclass classification error with respect to the number of parameters and the number of FLOPs per second during inference. The performance is evaluated on the SC35 testset. We compare the scaling of the hybrid network against its counterpart where the projection matrices are made complex.

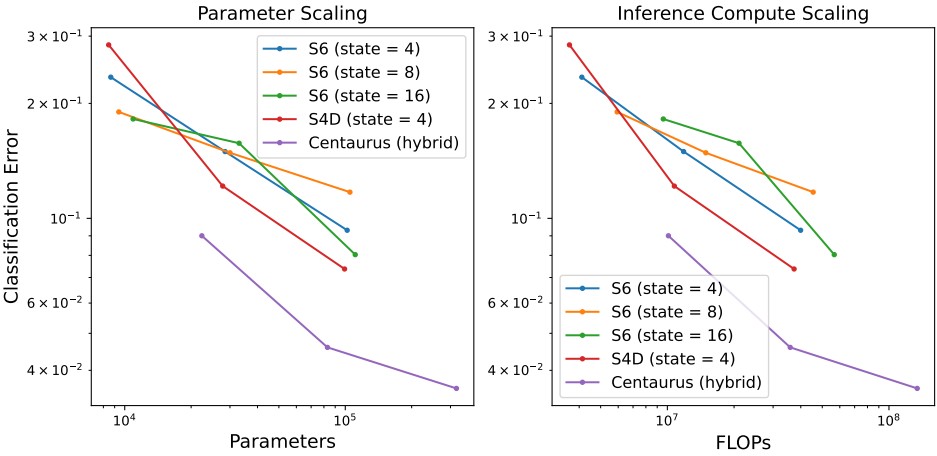

Figure 7: We compare the scaling of the hybrid network compared to homogeneous network variants where all SSM layers are S6 layers (or equivalently depthwise-separable SSM blocks where the SSM matrices are data-controlled).

Table 5: Comparing our Centaurus hybrid network against other deep state-space models on the SC10 testset. All networks operate on raw input waveforms sampled at 16000kHz. Note that only our Centaurus network performs downsampling, hence the small computational footprint (despite having more parameters). The numbers with an asterisk are estimated and not reported in the original works.

| Model | Accuracy ↑ | Parameters ↓ | FLOPs / sec ↓ |
|---|---|---|---|
| S4 (Gu et al., 2021) | 98.32 | 0.306M | 11.8G* |
| S4D-Lin (Gu et al., 2022) | 96.25 | 0.306M | 11.8G* |
| LiquidS4 (Hasani et al., 2022) | 98.51 | **0.224M** | - |
| S5 (Smith et al., 2022) | - | 0.280M | 13.5G* |
| **Centaurus** (hybrid) | **98.53** | 0.378M | **0.134G** |

Table 6: Resampling factor and output channel of each block of the network, which consists of an encoder performing down-samplings, an intermediate bottleneck, a decoder performing up-samplings, and finally an output processor. The number of sub-states for the "neck" SSM block is always 4.

| Layers | Resampling Factor | Channels | States |
|---|---|---|---|
| **Encoder** | | | |
| Block 1 (full) | 4 | 16 | 16 |
| Block 2 (full) | 4 | 32 | 4 |
| Block 3 (neck) | 2 | 64 | $128 \times 4$ |
| Block 4 (neck) | 2 | 96 | $128 \times 4$ |
| Block 5 (pw-neck) | 2 | 128 | 256 |
| Block 6 (pw-neck) | 2 | 256 | 256 |
| **Bottleneck** | | | |
| Block 1 (pw-neck) | 1 | 256 | 256 |
| Block 2 (pw-neck) | 1 | 256 | 256 |
| **Decoder** | | | |
| Block 1 (pw-neck) | 2 | 128 | 256 |
| Block 2 (pw-neck) | 2 | 96 | 256 |
| Block 3 (neck) | 2 | 64 | $128 \times 4$ |
| Block 4 (neck) | 2 | 32 | $128 \times 4$ |
| Block 5 (full) | 4 | 16 | 4 |
| Block 6 (full) | 4 | 1 | 16 |
| **Output** | | | |
| Block 1 (full) | 1 | 1 | 16 |
| Block 2 (full) | 1 | 1 | 16 |

the Audioset, Freesound, and DEMAND datasets. For all audio samples, we use the `librosa` library to resample them to 16 kHz and load them as numpy arrays.

For the LibriVox audio samples which form long continuous segments of human subjects reading from a book, we simply concatenate all the numpy arrays, and pad at the very end such that the array can be reshaped into (segments, $2^{17}$), or roughly 8.192 second segments. For all the other audio samples consisting of short disjoint segments, we perform intermediate paddings when necessary, to ensure a single recording does not span two rows in the final array. For audio samples longer than length $2^{17}$, we simply discard them. The input length to our network during training is then also $2^{17}$.

For every epoch, we use the entirety of the VCTK dataset and 10 percent of a randomly sampled subset of the LibriVox dataset. For each clean segment, we pair it with a randomly sampled noise segment (with replacement). The clean and noise samples are added together with an SNR sampled from -5 dB to 15 dB, and the synthesized noisy sample is then rescaled to a random level from -35 dB to -15 dB. Furthermore, we perform random temporal and frequency masking (part of the SpecAugment transform) on only the input noisy samples.

For all our trials in this experiment, we

- train for 500 epochs
- use a learning rate of 0.005 with a weight decay 0.02
- perform training on a single NVIDIA A30 GPU with a batch size of 192, resulting in around 1573 seconds of audio data per batch.

For the loss function, we combine SmoothL1Loss with spectral loss on the ERB scale, without any equalization procedure on the raw waveforms or spectrograms. The $\beta$ parameter of the SmoothL1Loss is set at 0.5, and the spectral loss is weighted by a factor that grows from 0 to 1 linearly during training.

See Table. 6 for the Centaurus network architecture. The SSM blocks themselves do not change the channel dimensions of the features. The downsampling and upsampling layers are trainable

convolutional layers, which at the same time perform channel projections. For example, if the input features are monoaural with shape (1, 16000), then the first SSM block will retain this shape, but the following downsampling block will project it to the shape (16, 4000). The network follows an hourglass architecture, with a long-range skip connection branching after every SSM block in the encoder, and merging before every corresponding SSM block in the decoder (i.e. output features of encoder block 1 is added to the input features of decoder block 6).

The homogeneous variants of the network is configured with the same architecture, with the same resampling factors and channels. To keep the parameters and FLOPs of the variants roughly at the same order, we use the following state configurations:

- For the depthwise-separable variants (S4D-like and S6-like), we use 64 states.
- For the pointwise bottleneck variant (S5-like), we use 256 states.
- For the bottleneck variant, we use 256 states, with 4 sub-states per state.
- For the full variant, we use 4 states per input-output channel pairing.

Surprisingly, using the Mamba blocks in their original form (including the highly complex nested macro-architecture) resulted in severe training plateaus, and resulted in many runs that did not outperform random guessing, despite much effort in hyper-parameter tuning and following the suggested "no weight decay" training recipe for the $A$ and $D$ matrices.

### E.3 SPEECH RECOGNITION

Table 7: Resampling factor, input/output channels, and internal states of each block of the Centaurus ASR network. The number of sub-states for the "neck" SSM block is always 4. The SSM block itself maintains the feature shape, followed by the downsampling layer projecting the temporal and channel dimension simultaneously. An optional feedforward or convolutional module can be additionally appended to the end of each block (again maintaining the feature shape). A block with a multiplier means that the block is stacked multiple times, with only the first occurrence of the block containing the downsampling layer (feature projection).

| Layers | Resampling Factor | In Channels | Out Channels | States |
|---|---|---|---|---|
| **Backbone (12 layers)** | | | | |
| Full Block | 5 | 1 | 16 | 64 |
| Full | 4 | 16 | 32 | 4 |
| Neck $\times 2$ | 2 | 32 | 64 | $128 \times 4$ |
| Neck $\times 2$ | 2 | 64 | 128 | $256 \times 4$ |
| PW-Neck $\times 2$ | 2 | 128 | 256 | 512 |
| PW-Neck $\times 4$ | 2 | 256 | 512 | 1024 |
| **Head (6 layers)** | | | | |
| PW-Neck | 4 | 512 | 512 | 1024 |
| PW-Neck | 2 | 512 | 512 | 1024 |
| PW-Neck $\times 4$ | 1 | 512 | 512 | 1024 |

The backbone is a 12-layer encoder, performing a total downsampling factor of 320, effectively streaming output features at a period of 20 ms. The output features from the encoder are then branched into two heads: 1) a smaller "character" head that is a 2-layer MLP, producing character logits, 2) and a larger "language" head, consisting of six additional pointwise-bottleneck SSM blocks, producing BERT token logits[13], additionally downsampling the features by a factor of 8. The character head is supervised with the soft logits from the `WAV2VEC2_ASR_LARGE_LV60K_960H` teacher model, and the language head is supervised via CTC loss with the tokenized transcripts. During inference, we simply extract the most probable token produced by the final layer of the "language head".

---

[13]Note that nowhere in our solution will we actually use the BERT network. We only use the BERT tokenizer for the token encoding of transcriptions and the token decoding of the network predictions.

The training pipeline involves supervised learning on transcribed audio samples. The training dataset includes the LibriSpeech full 960h training set, and the Multilingual LibriSpeech (MLS) English training set. The training lasts 100 epochs, with each epoch containing the full LibriSpeech training set and $1\%$ of the MLS English training set randomly sampled (so that a full training run will see all training data). To guarantee no data leakage, we check every sample in the training set and remove it if it is sufficiently similar to any sample in the development and testing sets. All the audio data are sampled at 16kHz before being fed into the network. Similar to the data pre-processing pipeline for speech denoising (see E.2), we pack the audio clips into segments of length 327680 (or roughly 20.5 seconds), introducing paddings when necessary to ensure a single recording does not span two segments.

For training the network, we

- train for 100 epochs
- use a learning rate of 0.02 with a weight decay of 0.1
- perform training with $8\times$ 40GB A100 with a batch size of 32 per GPU. This results in around 1.46 hours worth of audio data per effective batch.

The base Centaurus network can be augmented by appending an additional module after every SSM block, to promote more local modeling capabilities. We experiment with three types of (residual) modules here.

- Feed-forward network (FFN): This consists of two pointwise layers in the main path, making it purely feature mixing. Both layers preserve the channel dimension, with no expansion factors.
- Convolutional module (Conv): This is a residual module adapted from the Conformer architecture (Gulati et al., 2020). It consists of the pointwise-depthwise-pointwise pattern with an expansion ratio of 2 in the main path; the depthwise layer is a causal convolutional layer with a kernel size of 4.
- Mamba block: This is a complex residual gated block containing three nested skip connections, adapted from the Mamba macro-architecture (Gu & Dao, 2023). It begins by projecting the input channels into 4 times the original dimension, and splitting the channel features into two paths, one path for the core SSM operations, and another path for gating. The SSM layer itself contains an inner skip connection via the $D$ matrix (which we ignored for our Centaurus block). Additionally, there is an outer residual connection wrapping the entire Mamba block itself.

For using the Mamba block, we only applied it for the last 4 "language head" blocks as they are homogeneous. We also tested replacing the shallower blocks with Mamba blocks but it resulted in unstable training, which we believe was due to the highly complex Mamba blocks being too "close" to the less featurized raw audio signals. In addition, we also tested the Mamba blocks in their original form (with DWS gated S6 layers), but it also resulted in unstable training.

