# OpenReview forum: "Let SSMs be ConvNets: State-space Modeling with Optimal Tensor Contractions"
_ICLR.cc/2025/Conference — ICLR 2025 Spotlight_

### Official Review · Reviewer_HmD1 · 2024-10-25

**Soundness:** 4
**Presentation:** 3
**Contribution:** 3
**Rating:** 8
**Confidence:** 3

**Summary:**

This paper proposes an efficient deep state-space model, Centaurus. Inspired by traditional ConvNets, the authors explore a heterogeneous design using block variants similar to convolutional blocks. The model is framed through the perspective of Tensor Networks and optimized by ordering tensor contractions for efficiency. Experiments demonstrate that the model achieves competitive performance across multiple tasks, including keyword spotting, raw speech denoising, and ASR.

**Strengths:**

- The authors proposed a novel perspective by viewing state-space models (SSMs) as tensor networks, enabling efficiency optimizations through contraction order analysis. This tensor network approach is advantageous for visualization, allowing the authors to address efficiency as a congestion optimization problem on the base graph. The approaches demonstrated in the paper contributes a valuable exploration into efficient SSM models.
    - One of the optimizations involves applying FFT to transform the SSM into the frequency domain, allowing the optimal contraction order to be analyzed for improved efficiency. While FFT introduces a slight computational cost, which is not compute-bound, is minimal compared to the efficiency gains achieved through optimized contraction order.
    - The authors explore multiple variants of SSM operations and illustrate them in a figure, broadening the possibilities for combining heterogeneous blocks to enhance model performance. Earlier SSM models only used identical operations throughout.
    - The authors converted SSM into Frequency domain by FFT, which allows the optimization of determining the optimal “contraction order”. The paper also discussed the approach to evaluate the memory and computational requirements of an einsum contraction path involving FFT.
    - As an example, the author explained how they find the optimal contraction order of Bottleneck SSM Operation. The proof involved Lemma 1 that reveals the intermediate tensors have at most 3 dims, which restricted the potential optimization paths. This largely reduced the manual work of comparing different contraction orders.

- The proposed method demonstrates competitive performance with significantly enhanced efficiency. For keyword spotting, the model achieved the best accuracy with 100x fewer FLOPs. In raw speech denoising, the authors conducted an ablation study on different SSM variants and concluded that the hybrid Centaurus model achieved the highest PESQ score with the smallest parameter count and the fastest FLOPs/sec. For ASR, the model also showed competitive performance.

**Weaknesses:**

The paper includes a large amount of appendix material, likely due to page limitations. While the authors present key results in the main paper, leaving detailed proofs in the appendix, it would be beneficial to adjust the structure to highlight more of the original contributions and slightly condense other sections, such as the background. Although the background on SSM and einsum notation is important and well-explained, it occupies two full pages. For instance, a potential improvement could be briefly summarizing the proof of Lemma 1 and explaining how it restricts feasible patterns in the main paper. As it is an important detail of explaining how you explore the optimal order. It would reduce the need for readers to frequently switch between the appendix and the main text.

**Questions:**

- The expression of the depthwise SSM operation in Section 4.1, Figure 1, differs slightly from the expression in Section 4.2 (above Equation 9), particularly in the subindex. The expression in Section 4.2 includes a subindex “b” that is not present in the earlier section. Could you clarify the difference and explain the meaning of “b”? If it refers to the bit representation in quantization, it would be helpful to explicitly cover this detail in the paper.

- The authors provided an example of the most complex case—the bottleneck block—but skipped examples of other operations. Are there any general theories or principles that could be further distilled from these examples?

---

> ### Author Response · Authors · 2024-11-12
> **Paper restructure**
>
> We thank the reviewer for their useful suggestions! We will follow the suggestions and reorganize our paper such that the einsum exposition is more condensed, so we can spend more pages on providing a proof sketch of Lemma 1 in the main text and potentially allow us to walk through the contraction patterns for the other cases (full, grouped).
>
> The index b denotes "batch", and we will clarify this in our revisions. Thanks for catching this oversight! Batch size is actually an important factor in determining the optimal contraction order dynamically, so different batch sizes can yield different (but functionally equivalent) contraction paths during training.
>
> The general theory of optimizing tensor networks (einsum contractions) can actually be formulated as a minimal congestion problem (https://arxiv.org/abs/2002.01935) on general graphs, as we also cited in the main text. The original problem statement may need to be augmented to include details such as complex operations and FFT padding etc. Currently, our Centaurus model presents a contraction pattern with two sequence operands (input and kernel), but we are currently exploring the potential to introduce more than two sequences into the tensor network. We will likely leave this line of explorations to a future work (as it is not clear at this stage whether contractions involving more than two sequences can admit a natural recurrent form).

---

> ### Author Response · Authors · 2024-11-21
>
> We have submitted a revision incorporating the suggestions of the reviewer. We again thank the reviewer for making our manuscript better.
>
> > highlight more of the original contributions and slightly condense other sections, such as the background
>
> We have now condensed considerably the introduction and the exposition of the SSM layer definition and einsum notation to leave more room for our core contributions.
>
> > a potential improvement could be briefly summarizing the proof of Lemma 1
>
> We have added a small remark along with the "proof sketch" after Lemma 1 to present a minimal proof without needing the reader to switch to the Appendix.
>
> > Could you clarify the difference and explain the meaning of “b”?
>
> In the first paragraph of Section 4.2, we explicitly added that b represents batch size and noted the importance of considering batch size for determining the optimal contraction path.

---

> > ### Comment · Reviewer_HmD1 · 2024-11-22
> >
> > I appreciate the author's efforts for revising the paper and incorporating the suggestions provided. The revised version demonstrates improved readability and places greater emphasis on the core contributions, supported by a well-structured background introduction. I find the work to be solid and believe it presents a novel and meaningful contribution to the field.

---

### Official Review · Reviewer_x1nf · 2024-10-26

**Soundness:** 3
**Presentation:** 3
**Contribution:** 3
**Rating:** 8
**Confidence:** 3

**Summary:**

The authors present Centaurus, an adaption of state-space models (SSMs) / linear RNNs (lRNNs) to audio modeling / processing, where they embed these layers of different dimenionalities into a deeper hybrid architecture, where they take inspiration from classical CNN networks that use different strides, kernel sizes and feature dimensions at different levels in the network. Their ablations show that the hybrid composition is superior to homogenous depth-wise, bottleneck and point-wise bottleneck architectures, and their models show strong performance while being superiorly FLOP-efficient compared to the current SOTA. Theoretically, they connect their hybrid architecture adaptions to tensor networks combined with Fourier operations and show the optimal tensor contraction strategy for their network.

**Strengths:**

Their hybrid architecture shows better results at lower FLOPs, while retaining a similar scaling behaviour compared to other homogeneous models across model sizes.

**Weaknesses:**

The proposed model is a combination of known primitives (SSMs and inhomogeneous scaling from CNN architectures).
The work does not cite and compare to other relevant, related work that potentially outperforms the trained models on the given datasets, e.g. Zhao et al. 2022: "Monaural speech enhancement with complex convolutional block attention module and joint time frequency losses." for the VB-DMD dataset.

**Questions:**

Since both RNNs and State Space models are well-known, a detailed introduction of these architectures can be omitted, except for the parallelization / transformations to/from the Fourier domain.
Einsum notation is definitely superior for Tensor expressions, so also here the extensive introduction could be reduced.

---

> ### Author Response · Authors · 2024-11-13
> **DCUnet and FRCRN**
>
> We are glad to hear that the reviewer appreciates our hybrid network. We would like to address several comments the reviewer raised.
>
> > The proposed model is a combination of known primitives (SSMs and inhomogeneous scaling from CNN architectures).
>
> We believe that our work serves more than a combination of SSM and CNN primitives. The core innovation, as the reviewer identified, is the reformulation of SSMs as tensor networks (einsum contractions) in the frequency domain, such that the connective structures of CNN blocks can be applied. Without this tensor contraction structure, we believe it would be much more difficult to unveil the connection between SSM and CNN blocks.
>
> On the algorithmic end, the concept of "optimal tensor contraction" is (arguably) meaningful only for SSM blocks, and is not explored too much for classical CNNs. For classical CNNs (e.g. 2d convnets), the convolution kernel is finite-window and explicitly parameterized, meaning that the convolution order is essentially "forced" to be performed sequentially layer by layer (even in the absence of non-linear activation functions). For SSMs however, the convolution kernel is infinite-window (IIR) and additionally parameterized by state matrices. This means that we have the freedom to move between temporal and frequency domains, and the freedom to choose whether to "materialize" the internal states or not. These are not features allowed by classical CNNs, as there are no obvious concepts of "states", and performing FFTs to do finite-window convolutions would be far too inefficient.
>
> > related work that potentially outperforms the trained models on the given datasets, e.g. Zhao et al. 2022:
>
> Regarding this baseline, we actually came across the two models by the same group: [DCUnet](https://arxiv.org/abs/2102.01993) and [FRCRN](https://arxiv.org/abs/2206.07293), one of which the reviewer mentioned. We will make appropriate references to these works in our main text.
>
> For the DCUnet model, we could not find an official implementation of it, and the DEMAND dataset reported (dataset 1) is mixed with WSJ0, but we reported on the more standard VCTK + DEMAND instead (VB-DMD). The WSJ0 is usually an easier benchmark, so we do not think a comparison would be fair. In addition, their MC variant (using CCBAM module) is a non-causal model due to the use of a global average pooling (GAP) operation. We are more focused on comparing against real-time solutions instead.
>
> The FRCRN model, however, we do believe is a real-time network, and their performance on the VB-DMD testset is a PESQ of 3.21, which is 0.04 points short of our best-performing model. We have experimented with this model and recalled it having significantly more parameters and FLOPs requirements. We will revisit this and double-check everything, and include this in our Table 1 if appropriate. We thank the reviewer for bringing this up!
>
> We would be happy to address any additional questions or comments that the reviewer may have.

---

> > ### Comment · Reviewer_x1nf · 2024-11-21
> > **Table 1 - Performance Comparison**
> >
> > Thank you for your response. Regarding Table 1, the choice of the baselines is still unclear to me. As is, the table implies that Centaurus beats all baselines in performance, while having lower computational demands and parameters. However, stronger baselines exist - with larger computational demand. It would be nice to clarify and justify the choice of baselines in the paper, for these tables to be more solid.
> >
> > Otherwise, I think this work is definitely valid and good contribution.

---

> ### Author Response · Authors · 2024-11-22
>
> We have submitted a revision incorporating the suggestions of the reviewer. We again thank the reviewer for making our manuscript better.
>
> >  The work does not cite and compare to other relevant, related work that potentially outperforms the trained models on the given datasets, e.g. Zhao et al. 2022
>
> We have now included the FRCRN model, which is a model related to the suggested reference, that we could fairly compare against (PESQ on VB-DMD). This is added to the first row of Table 2.
>
> >  the table implies that Centaurus beats all baselines in performance, while having lower computational demands and parameters
>
> We acknowledged that there may be networks of larger sizes that can outperform the Centaurus network, and this is now stated explicitly at the beginning of page 10. However we still note that those models are mostly offline (non real-time) networks, so we did not include them in Table 1 due to tight space constraints. To name the prime example, we can look at the SEMamba network as it is directly relevant to our network
>
> |               | PESQ   | params   | FLOPs   |
> |---------------|------------|------------|------------|
> | **SEMamba** (online, causal)     |   2.76    |      3.6M     |    0.76G        |
> | **SEMamba** (offline, bi-directional)     |       3.52     |      2.25M      |      65.46G      |
> | **Centaurus** (online, with causal conv)     |      3.25      |      0.51M      |      0.29G      |
>
> An interesting future direction would be then to study a bi-directional variant of the Centaurus network, so they could compete with other offline speech enhancement and ASR networks. This extension should also be encompassed by our optimal tensor contraction formalism (with the FFT convolution structure modified slightly)
>
> Other examples of models outperforming Centaurus, but being offline are: **MP-SEnet** using a frequency transformer, **SCP-GAN** using a time-frequency conformed and a Bi-LSTM generator, **D2Net** using a global-local dual-path transformer

---

> > ### Comment · Reviewer_x1nf · 2024-11-24
> >
> > Thank you for the clarifications and the update in the paper. Now the comparison is more comprehensive.

---

### Official Review · Reviewer_j8MZ · 2024-11-05

**Soundness:** 3
**Presentation:** 3
**Contribution:** 3
**Rating:** 8
**Confidence:** 3

**Summary:**

The authors present an approach to augment the number of features in the inputs, outputs and internal states of LTI systems, leveraging state space models, in order to facilitate architectural flexibility towards building abstractions that vary feature dimensions in neural models, such as those in CNNs. The authors present connections between tensor networks, which generalize low-rank decompositions in a way that can be applied to SSM einsums.

The authors introduce a weighting for basis kernels which can be adapted to a recurrent system. Convolutions, in this way, are shown as sums of various real and complex combinations of Fourier modes. The authors use this idea to present Centaurus, a configuration of tensor networks that allows retains SSM blocks but projects internal state to different sizes, making it possible to mimic various CNN abstractions such as depthwise, grouped, and bottleneck convolutions.

The authors construct each of these layers and present implementation details, i.e. a tradeoff between optimization of einsum expressions (for which we have few compiler tools at the moment) versus GPU kernel customization.
 The authors present experiments on three tasks within sequence modeling/speech: keyword spotting, raw speech denoising, and speech transcription, showing strong results in each area with small models with good computational scaling properties.

**Strengths:**

- The presentation of methodology is convincing and strongly tied to first principle sin SSMs.
- The application of tensor networks to solve the problem of projecting kernel matrices to and from frequency spaces of different dimensions is novel and elegant.
- The presentation of implementation and computational considerations shows care was given to scaling tradeoffs, i.e. memory-boundedness of this regime of compute regime, opportunities (or lack thereof) for operator fusion, and operation/contraction ordering per kernel construction.
- Results are compelling — in particular, efficiency of the proposed models is impressive given performance.
- Attention to detail with the model name is nice.

**Weaknesses:**

- Baselines could be much stronger, in that there are no other SSM model baselines. Ablations are mostly within the architectural innovations present with Centaurus.
- The constraint that some projection matrices from basis kernels must be real constrains the expressiveness of the model, although it’s likely that such generalizations in future work might enable this.
- A more comprehensive architectural ablation would make the paper stronger.  While there are explanations of architectures in appendix E, there is no study or comparison of components to CNNs. Do architectural ideas useful in building CNNs apply to building Centaurus models?

**Questions:**

- The authors should cite other work in speech that explores other convolutional architectures, such as ([Kirman et al.](https://ieeexplore.ieee.org/iel7/9040208/9052899/09053889.pdf), time-depthwise separable convolutions, [Hannun et al.](https://arxiv.org/pdf/1904.02619), or other extensive architectural ablations based on computational efficiency for training and inference: [Pratap et al.](https://arxiv.org/pdf/2001.0972))
- ASR baselines can examine other convolutional architectures such as those presented in [Synnaeve et al.](https://arxiv.org/pdf/1911.08460); these baselines are slightly better comparisons given that they are also end-to-end and also present results language-model based decoding.
- Experiments in Appendix E leverage samples from Libri-Vox — was they Libri-Light or samples of raw data?
- A brief discussion of future work/applications to other domains (outside of speech) for CNNs (such as computer vision tasks) might strengthen the paper.

---

> ### Author Response · Authors · 2024-11-12
> **Strong baselines, complex projections, and more CNN ideas**
>
> We thank the reviewer for their comments and pointers to the relevant literature. We will incorporate the appropriate ASR baselines as the reviewer suggested into our discussion and the Table.
>
> 1. For the speech commands dataset (SC35), our goal is mainly for self-ablations to highlight the usefulness of a hybrid SSM design. In Table 5 on page 20 (Appendix E.2), we compared against previous "SOTA" models like S4D, S5, and Liquid-S4 in their original form, and showed an 100x reduction in inference FLOPs while achieving the best accuracy
>
> 2. For raw audio denoising, we are mostly interested in real-time denoising solutions, with DeepFilterNet3 being a very strong baseline. However, we may have missed other relevant works in this area (as also pointed out by reviewer x1nf), so we will do a broader review on that.
>
> Regarding the complexity of the B and C matrices, the reviewer is correct that restricting them to real will incur loss of generality, as we also commented on line 156 of the main text. However, in practice, we decided to keep them real for the reasons that: 1) complex numbers are still inefficiently supported by CUDA, 2) our empirical studies showed that complex projection matrices did not yield a significant gain in performance. We will try to perform a more systematic study in that regard, and report the results here or in the Appendix of our updated revision.
>
> The reviewer is correct that there are in fact more "CNN ideas" that we have not explored. In this work, we are mainly focused on "connective structure" of CNNs which is enabled by our general contraction patterns. Another general idea that comes to mind is "stride 2" convolution versus "stride 1 with downsampling", the latter of which is being performed by Centaurus (and most other SSM networks). This may be an interesting direction to explore, as stride 2 convolutions may require non-trivial modifications to the FFT convolution algorithm (i.e. using a polyphase filter). We will see if we can produce anything useful along this idea within the review time frame. (A small sidenote, our denoising network uses an "hourglass" structure, which is a common structure in CNN-based networks in computer vision, such as object detection. However, this is not novel and was originally proposed in the Sashimi audio generation network).
>
> We again thank the reviewer for their suggested literature, and will appropriately include them in our revisions. For our training data of the raw audio denoising network, we downloaded the clean samples from the DNS4 challenge (https://github.com/microsoft/DNS-Challenge/blob/master/download-dns-challenge-4.sh), which is a superset of the DNS1 challenge, hence including LibriVox samples that are filtered and cleaned by Microsoft (which we believe is likely not the same as LibriLight).

---

> ### Author Response · Authors · 2024-11-22
>
> We have submitted a revision incorporating the suggestions of the reviewer. We again thank the reviewer for making our manuscript better.
>
> > The constraint that some projection matrices from basis kernels must be real constrains the expressiveness of the model, although it’s likely that such generalizations in future work might enable this.
>
> We have included a study that makes the projection matrices (B, C, and E) complex, similar to the original implementation of S4D and S5. Surprisingly, it did not appear to outperform the real variant, so it appears that real projection matrices should be sufficient for deep SSMs, as echoed by the Mamba paper. The results are presented in Fig. 6 in Appendix E.1
>
> > Do architectural ideas useful in building CNNs apply to building Centaurus models? A brief discussion of future work/applications to other domains (outside of speech) for CNNs (such as computer vision tasks) might strengthen the paper.
>
> In the conclusion Section, we have added a future direction to study "convolutional structures in higher dimensions, incorporating additional structures such as stride, dilation, and transposition, to see whether Centaurus can be effective for computer vision tasks as well". We are currently working on such studies, but it does not appear we can complete them in the very short-term. We again thank the reviewer for this inspiration.
>
> > The authors should cite other work in speech that explores other convolutional architectures
>
> We have added these references now in Section 1 and Section 2.2
>
> > ASR baselines can examine other convolutional architectures such as those presented in Synnaeve et al.
>
> We have added the "fully conv" architecture in the suggested literature, to Table 3 of our manuscript

---

> > ### Comment · Reviewer_j8MZ · 2024-12-02
> >
> > No additional questions — thanking the authors for the additional details and proposed modifications to the manuscript. I leave my score unchanged.

---

### Official Review · Reviewer_Td3n · 2024-11-05

**Soundness:** 3
**Presentation:** 2
**Contribution:** 2
**Rating:** 6
**Confidence:** 3

**Summary:**

The paper presents a series of hybrid state-space models (SSMs), inspired by the design of convnets. It achieves this by generalizing 1D kernels to 2D and designing the depthwise, group, and full convolution counterparts for SSMs. Experiments on several sequence modeling tasks validate the effectiveness of the proposed method.

**Strengths:**

1. The concept of generalized SSM blocks, configurable with flexible connectivity structures, demonstrates good soundness and could complement existing SSM designs.

2. The proposed method is supported by both theoretical analysis and empirical experiments.

**Weaknesses:**

1. The major concern is that the real-device training/inference efficiency of the proposed generalized SSM blocks is not provided. It is unclear whether the flexibility of these building blocks comes at the cost of reduced real-device efficiency.

2. Another major concern is that the position of the proposed method among the latest SOTA SSMs is unclear. For example, Mamba has introduced input adaptivity in their selective state design and employed an advanced macro-structure with gating mechanisms and FFNs to enhance channel mixing in addition to token mixing. These advancements also address the limited expressiveness of vanilla SSMs, which are mostly depthwise-separable. The authors are expected to demonstrate whether the proposed method is a better choice or is orthogonal and can be combined with these advanced designs.

3. The proposed method is evaluated on small-scale sequence modeling tasks. It would be highly desirable for it to be evaluated on language modeling and commonsense reasoning tasks.

4. The writing clarity could be improved by better explaining certain terms. For example, the concept of "depthwise-separable" in the context of SSMs is important to the logic of this paper and should be better clarified.

**Questions:**

My concerns are listed in the weaknesses section.

I have one additional question: For the generalized SSM blocks with group/full convolution patterns, can I understand that these blocks integrate both token-mixing and channel-mixing within a single block, making other channel-mixing blocks, such as FFNs, unnecessary?

---

> ### Author Response · Authors · 2024-11-12
> **Data gating, language modeling, and channel mixing**
>
> We thank the reviewer for their insightful comments. We begin by addressing the concern of interfacing with networks using data-gating (e.g. Mamba, GLA, HGRN2). We believe our innovation of general contraction (mixing) pattern can be viewed as being orthogonal to data-gating, and can be in theory integrated with this mechanism. However, this may present some interesting algorithmic and engineering challenges, which actually makes it an important future direction to pursue.
>
> The core challenge is that under a dynamic state evolution matrix (A matrix), the internal states may need to be explicitly materialized (both in theory and practice), so this by itself will limit the freedom of contraction. Furthermore, the dynamicity of the A matrix makes the underlying kernel (or Green's function to be pedantic) non-time-invariant, which makes the FFT conv technique not applicable. However, there are recent techniques using state-passing that can formulate such systems as a series of tensor contractions (e.g. GLA and Mamba2); the drawback here is that a new operand index L' is needed, so the dimensionality of the A matrix needs to be reduced (or "shared A elements" in Mamba2) such that no large intermediate tensors are produced. This actually echos Lemma 1 of our paper, so there may be some underlying theory that can motivate the use of a more optimal tensor contraction pattern for data-gated SSM layers. At a very high level, we believe there is a tradeoff between connective structure and data-gating, with Centaurus being on one end (optimized contraction patterns with no data-gating), and Mamba being on the other (simple contraction with data-gating). It is likely the optimum is somewhere in the middle.
>
> On the empirical end, we will work on trying to improve our empirical section by incorporating the selective scan layer (S6, or the core mamba layer) into our ablations and comparisons as well, so we get a fuller comparison in the SSM landscape. Currently, the runs on the ASR experiments are already nearing the limit of our current computing resources, and it may be difficult to perform extensive experiments on tasks such as large language modeling in the very short term; however, we do have plans to explore this in the near future. Our intuition here is that language models lend themselves to a rather homogenous structure, due to the nature of the task (not requiring downsampling), so it is unclear at this stage whether a heterogenous design would offer considerable improvements. However, it is still possible to use a general Centaurus block with some underlying tensor network structure (e.g. Tucker decomposition), and repeat it multiple times like the transformer model. The optimized contraction pattern can compensate for the lack of data-gating mechanism, or alternatively be integrated with data-gating itself, as suggested above.
>
> In regards to the comment on real-device inference, the reviewer is correct that there may be a tradeoff where a heterogeneous network may be less efficiently supported, especially on specialized hardware. We believe this is less of a problem on standard CPU or GPU hardware, as we still maintain the modularity of each SSM block. A general concern may be that during inference, the internal states may need to be maintained and updated in the SRAM (or some form of quick but small memory) due to frequent accesses, hence why a main focus of this paper is to structure our networks such that they result in small internal state banks during inference.
>
> We thank the reviewer again for suggesting improvements to our writing clarity, and we will revise our paper accordingly. Regards to the question about channel mixing, the reviewer is absolutely correct that the channel-mixing mechanism is baked into our model (like S5 and Mamba2), so that FFN blocks (or additional linear projection layers) are not required. In fact, most of our experiments are performed without additional FFN blocks. But for certain tasks (like ASR), adding additional FFN blocks can help performance. (In Mamba1/2, the input and output projection layers effectively serve as FNNs as well)
>
> (As a small sidenote, unlike the S4ND or Simba networks, our SSM kernels are still 1D kernels. The core inspiration from CNN is their connective structure (depthwise, group, full, as the reviewer pointed out), rather than their spatio-temporal dimensionality. However, it does also seem possible to integrate our framework with high dimensional state-space models (1d to 3d), and this itself may also be an interesting future direction of study.)

---

> ### Author Response · Authors · 2024-11-22
>
> We have submitted a revision incorporating the suggestions of the reviewer. We again thank the reviewer for making our manuscript better.
>
> > The major concern is that the real-device training/inference efficiency of the proposed generalized SSM blocks is not provided. It is unclear whether the flexibility of these building blocks comes at the cost of reduced real-device efficiency.
>
> We have now included a training benchmarking study on an A100 GPU comparing our optimized tensor contraction designed for our generalized blocks against a naive contraction pattern used in previous works (e.g. S4D and S5). The results are presented in Fig. 4 of Appendix C (with the benchmarking details there as well). Here is a snippet of the scalability study, where the time represents one step of forward plus backward pass, where naive contraction ran into VRAM issues in the end.
>
> State scaling:
> |               | N=32  | N=64  | N=128 | N=256 | N=512 | N=1024 | N=2048 |
> |---------------|------------|------------|------------|------------|------------|------------|------------|
> | naive contraction (ms)  |     6.05   |   11.71   |   23.9   |  47.8 | 96.5 | - |  - |
> | optimized contraction (ms) |  4.96 | 5.05 | 5.35 | 5.94 | 7.18 | 9. 62 | 14.5 |
>
> Length scaling:
> |               | L=32  | L=64  | L=128 | L=256 | L=512 | L=1024 | L=2048 |
> |---------------|------------|------------|------------|------------|------------|------------|------------|
> | naive contraction (ms)  |  1.71 | 1.69 | 2.84 | 5.64 | 11.6 | - | - |
> | optimized contraction (ms) | 1.40 | 1.40 | 1.40 | 1.40 | 1.50 | 3.04 | 6.48 |
>
> > Another major concern is that the position of the proposed method among the latest SOTA SSMs is unclear.
>
> We have now included additional studies in all three experiments (keyword spotting, denoising, ASR) to include comparisons against both the S6 layer itself (selective scan) and the Mamba macro-architecture (three nested skips and silu gating). At a high level, we found neither S6 nor the Mamba macro-block was strictly required to improve the performance on raw audio tasks. This is most likely due to the nature of the task being interfaced with less featurized data (raw audio), so the selective scan mechanism or complicated residual gating may not be too helpful (though they do not seem to hurt performance either).
>
> In Fig. 3 and Fig. 7, we compare Centaurus and S4D against the S6 layer with different numbers of states (4, 8, and 16), and found that it didn't offer a major improvement over the vanilla S4D. In addition, for raw audio denoising, having S6 layers also did not help with the training plateau phenomenon that S4D suffered from. For ASR, we included a variant of the Centaurus network using the Mamba macro-architecture, and the performance did slightly improve, but still on par with the other variants if accounted for parameters and FLOPs. This is most likely because both the Mamba macro-architecture and the convolutional module contain a causal depthwise conv and two linear projection layers (just arranged and connected differently), and it is not clear that the specific macro-gating structure of Mamba is the superior one for audio tasks.
>
> Note that here we are not extending our conclusions to other tasks such as language modeling, where the input features are more "discrete" compared to more "continuous" data such as audio. It is entirely possible that selective scan and the Mamba macro-architecture are important for "discrete" data (and proven so in the original Mamba paper). Thus in these "discrete-data" domains, it may be beneficial to then interface Centaurus with such "Mamba features".
>
> > It would be highly desirable for it to be evaluated on language modeling and commonsense reasoning tasks.
>
> We have acknowledged in the conclusion that language modeling is an important future direction, and has already begun some initial studies on it. We do not have anything concrete to report in the paper yet, but initial studies using a ~1B tensorized Centaurus network training on SlimPajama show promising results compared to Mamba, GLA, and HGRN2. We do not discuss this extensively here as it is out of scope.
>
> > the concept of "depthwise-separable" in the context of SSMs is important to the logic of this paper and should be better clarified.
>
> We thank the reviewer for catching this unclarity. In Section 4.1, we have now included an extended discussion of depthwise-separability in the context of SSM blocks, and contrasted it with a "full" SSM operation. In particular, we pointed out that a DWS block does not contain channel mixing, so an additional channel mixing (pointwise) layer should be appended.

---

> > ### Comment · Reviewer_Td3n · 2024-11-26
> >
> > Thank you to the authors for preparing the extensive response! I will raise my score. I hope the clarity of this work can be further improved in the final version.

---

> > > ### Author Response · Authors · 2024-11-26
> > >
> > > We are glad to hear the reviewer found our responses helpful, and thank you again for your insightful comments!
> > >
> > > We will continue to improve on the clarity of our work until the final version. Also happy to continue our discussion (if needed) through the extended discussion period.

---

### Author Response · Authors · 2024-11-22
**Revision highlights**

Dear reviewers and area chairs, we thank everyone for this review process, and have submitted a revision addressing all the feedback that the reviewers have provided. Here is a summary of the core changes.

We have condensed Sections 1 and 2, while at the same time including additional references pointed out by the reviewers. The exposition of SSMs and einsum notation (Sections 3.1 and 3.2) has also been condensed, but keeping the core concepts. In Section 3.3, we have included an extended discussion comparing the Centaurus network with data-controlled SSMs (particularly Mamba), and discussed how they are different and can potentially be interfaced with each other.

In Section 4.1, we provided a discussion of depthwise-separability and compared it against a "full" SSM layer, in the context of channel mixing. In Section 4.2, we discussed the importance of batch size in the dynamic determination of the optimal contraction order. In Section 4.3, we added a proof sketch of Lemma 1, showing how it restricts the feasible contraction paths.

In Section 5.1 (keyword spotting), we have added the S6 layer as an additional comparison. In Section 5.2 (speech enhancement), we added **FRCRN** and **SEMamba** as additional baselines, and the S6 layer into our ablations as well. In Section 5.3 (ASR), we added a fully convolutional network as a baseline, and additionally included the Mamba macro-block into our ablations.

In the conclusion section, we discussed future directions related to data-control mechanism (for language modeling) and additional ConvNet concepts (for computer vision)

In the Appendix C, we added benchmarking for our optimized contraction algorithm on an A100. In Appendix D, we additional discussed the estimation of parameters and FLOPs for complex projection matrices, for the selective-scan mechanism, and for the Mamba macro-architecture. In Appendix E.1 (keyword spotting), we performed additional studies with complex projection matrices and with the S6 layers of different state dimensions. In Appendix E.3 (ASR), we discussed the details on how the Mamba macro-architecture is integrated into the "language head" of the Centaurus ASR network.

We believe our manuscript is now much stronger thanks to the review process, and we believe that our work will be a novel and important contribution to deep state-space modeling.

---

### Meta-Review · Area_Chair_eMEV · 2024-12-15

**Metareview:**

The paper presents an innovative approach to state-space modeling inspired by convnets. The methodology is convincing and results are compelling, especially in speech-related tasks. In the review and discussion process, weaknesses include lack of clear real-device efficiency data initially, uncertainty in position among other SSMs, weak baselines, model expressiveness constraints, lack of comprehensive architectural ablation, and writing clarity issues. The authors have made significant efforts to address reviewer comments in revisions, improving the clarity and strength of the paper. Overall, the work shows promise with unanimous acceptance scores.

**Additional Comments On Reviewer Discussion:**

The reviewers mainly raised concerns about real-device efficiency, SOTA comparison, model expressiveness, ablation studies, and writing clarity. The authors responded to these concerns and provided a revised version of paper. The main concerns have been well addressed in the discussion.

---

### Decision · Program_Chairs · 2025-01-22

Accept (Spotlight)